# Hypoxia-inducible factor 1 signaling drives placental aging and can provoke preterm labor

Erin J Ciampa[1]*, Padraich Flahardy[1], Harini Srinivasan[2], Christopher Jacobs[2], Linus Tsai[2], S Ananth Karumanchi[3], Samir M Parikh[4]*

[1]Department of Anesthesia, Critical Care, and Pain Medicine, Beth Israel Deaconess Medical Center, Harvard Medical School, Boston, United States; [2]Division of Endocrinology, Beth Israel Deaconess Medical Center, Harvard Medical School, Boston, United States; [3]Department of Medicine, Cedars-Sinai Medical Center, Los Angeles, United States; [4]Division of Nephrology, Departments of Internal Medicine and Pharmacology, University of Texas Southwestern Medical School, Dallas, United States

*For correspondence:
eciampa@bidmc.harvard.edu
(EJC);
Samir.Parikh@utsouthwestern.
edu (SMP)

**Competing interest:** The authors declare that no competing interests exist.

**Abstract** Most cases of preterm labor have unknown cause, and the burden of preterm birth is immense. Placental aging has been proposed to promote labor onset, but specific mechanisms remain elusive. We report findings stemming from unbiased transcriptomic analysis of mouse placenta, which revealed that hypoxia-inducible factor 1 (HIF-1) stabilization is a hallmark of advanced gestational timepoints, accompanied by mitochondrial dysregulation and cellular senescence; we detected similar effects in aging human placenta. In parallel in primary mouse trophoblasts and human choriocarcinoma cells, we modeled HIF-1 induction and demonstrated resultant mitochondrial dysfunction and cellular senescence. Transcriptomic analysis revealed that HIF-1 stabilization recapitulated gene signatures observed in aged placenta. Further, conditioned media from trophoblasts following HIF-1 induction promoted contractility in immortalized uterine myocytes, suggesting a mechanism by which the aging placenta may drive the transition from uterine quiescence to contractility at the onset of labor. Finally, pharmacological induction of HIF-1 via intraperitoneal administration of dimethyloxalyl glycine (DMOG) to pregnant mice caused preterm labor. These results provide clear evidence for placental aging in normal pregnancy, and demonstrate how HIF-1 signaling in late gestation may be a causal determinant of the mitochondrial dysfunction and senescence observed within the trophoblast as well as a trigger for uterine contraction.

## eLife assessment

This **valuable** study provides insights into mechanisms of placental aging and its relationship to labor initiation. The authors provide **solid** evidence and have thoroughly investigated the molecular characteristics of normal placental aging using in vivo and in vitro model systems and human placental tissue analysis to corroborate their findings. This work contributes to existing work in placental aging and preterm birth and will be of interest to reproductive scientists.

## Introduction

Preterm birth (birth prior to 37 completed weeks of gestation) is a massive global health burden: it leads all causes of death in neonates and children to the age of 5 worldwide (*Liu et al., 2015*), and survivors face a broad array of short- and long-term health challenges (*Blencowe et al., 2013*). Most

preterm births in the United States are due to spontaneous onset of preterm labor, with unknown underlying cause (*Goldenberg et al., 2008*). Preterm labor has proven difficult to treat; there currently exist no highly effective interventions that prevent spontaneous preterm birth (SPTB) (*Smith et al., 2009*; *Romero et al., 2014*). The dearth of effective treatments stems from our lack of understanding about the pathways regulating spontaneous onset of labor, both preterm and at term.

The placenta defines the maternal-fetal interface; it is capable of profoundly influencing both maternal and fetal physiology (*Kiserud et al., 2006*; *Shaut et al., 2008*; *Lykke et al., 2009*), and it is subject to remodeling in response to its local environment (*Genbacev et al., 1996*). Placental disease is known to contribute to other disorders of pregnancy including preeclampsia (*Plaks et al., 2013*; *Li et al., 2016*; *Romero and Chaiworapongsa, 2013*) and intrauterine growth restriction (*McIntyre et al., 2020*; *Xu et al., 2021*) and there is increasing interest in its potential role as a driver of preterm labor (*Koga et al., 2009*; *Pique-Regi et al., 2019*; *Beharier et al., 2020*).

Recent studies have found that placentas from pregnancies that resulted in SPTB have unique metabolomic and transcriptomic signatures from term counterparts (*Elshenawy et al., 2020*; *Lien et al., 2021*; *Paquette et al., 2018*). The reported differences broadly implicate the stress response, inflammation, and various metabolic pathways. It has been challenging to narrow these findings or translate them for mechanistic relevance, given the lack of suitable culture systems that integrate experimental manipulation of placental cells with the uterine myocyte response. Furthermore, inter-pretation studies profiling human placentas from SPTB is sharply limited by the lack of gestational age-matched controls. Without these, the effects of gestational age cannot be distinguished from factors driving premature labor. The placenta encounters a dynamic local environment across its life-time and faces evolving needs of the growing fetus, so the context of normal gestational age-related changes is vital for the correct interpretation of molecular characteristics distinguishing an SPTB placenta. Defining the molecular-level changes in the *healthy* placenta as it approaches the end of gestation, and their potential effects on the timing of labor onset would therefore address important knowledge gaps.

Here, we report findings from unbiased transcriptomic analysis of healthy mouse placenta, which highlighted hypoxia-inducible factor 1 (HIF-1) signaling as a hallmark of advanced gestational time-points, accompanied by mitochondrial dysfunction and cellular senescence. We detected some similar effects in human placentas, then modeled HIF-1 induction with two different stimuli and in two trophoblast cell models, demonstrating that mitochondrial dysfunction and cellular senescence arise secondary to HIF-1 stabilization. Whole transcriptome analysis revealed that upon HIF-1 stabili-zation, a trophoblast cell line acquires signatures that recapitulate the aged placenta. Finally, we show that conditioned media from these cells is sufficient to potentiate a contractile phenotype in uterine myocytes, implicating a mechanism by which the aging placenta may help drive the transition from uterine quiescence to contractility at the onset of labor. This mechanism is further reinforced by an in vivo model in which administration of the prolyl hydroxylase inhibitor dimethyloxalyl glycine (DMOG) to pregnant mice induces HIF-1 signaling in the placenta and causes SPTB.

## Results

To illuminate gestational age-dependent transcriptional changes in across a healthy pregnancy, we collected whole mouse placentas at 48 hr intervals spanning embryonic day 13.5–17.5 (e13.5–e17.5) and quantified mRNA and protein targets that emerged from network analysis of an independently published microarray dataset from healthy mouse pregnancies. Upon searching Gene Expression Omnibus (GEO) (*Edgar et al., 2002*) datasets for 'placenta AND transcriptome', we were surprised that among 326 results, only 9 included data from normal placenta across a series of gestational time-points extending to late pregnancy (*Knox and Baker, 2008*; *Zhou et al., 2009*; *Loux et al., 2019*; *Soncin et al., 2018*; *Maeda et al., 2019*; *Morey et al., 2021*; *Steinhauser et al., 2021*; *Figure 1*). We applied weighted gene correlation network analysis (WGCNA) to a microarray study of mouse placenta (*Knox and Baker, 2008*) (GEO accession GSE11224) spanning e8.5 to postnatal day 0 (p0) to assess the mRNA signature of the aging mouse placenta, using the dataset with the best temporal resolu-tion across gestation. Distinct clusters emerged, each reflecting a group of genes whose expression changes across timepoints in a unified way (*Figure 2A*; accompanying statistics available at Mendeley Data, doi: 10.17632/g6vrw9jjn4.1). Genes in the 'blue' cluster showed increasingly positive correla-tion with advancing gestational timepoints; KEGG functional pathways significantly overrepresented

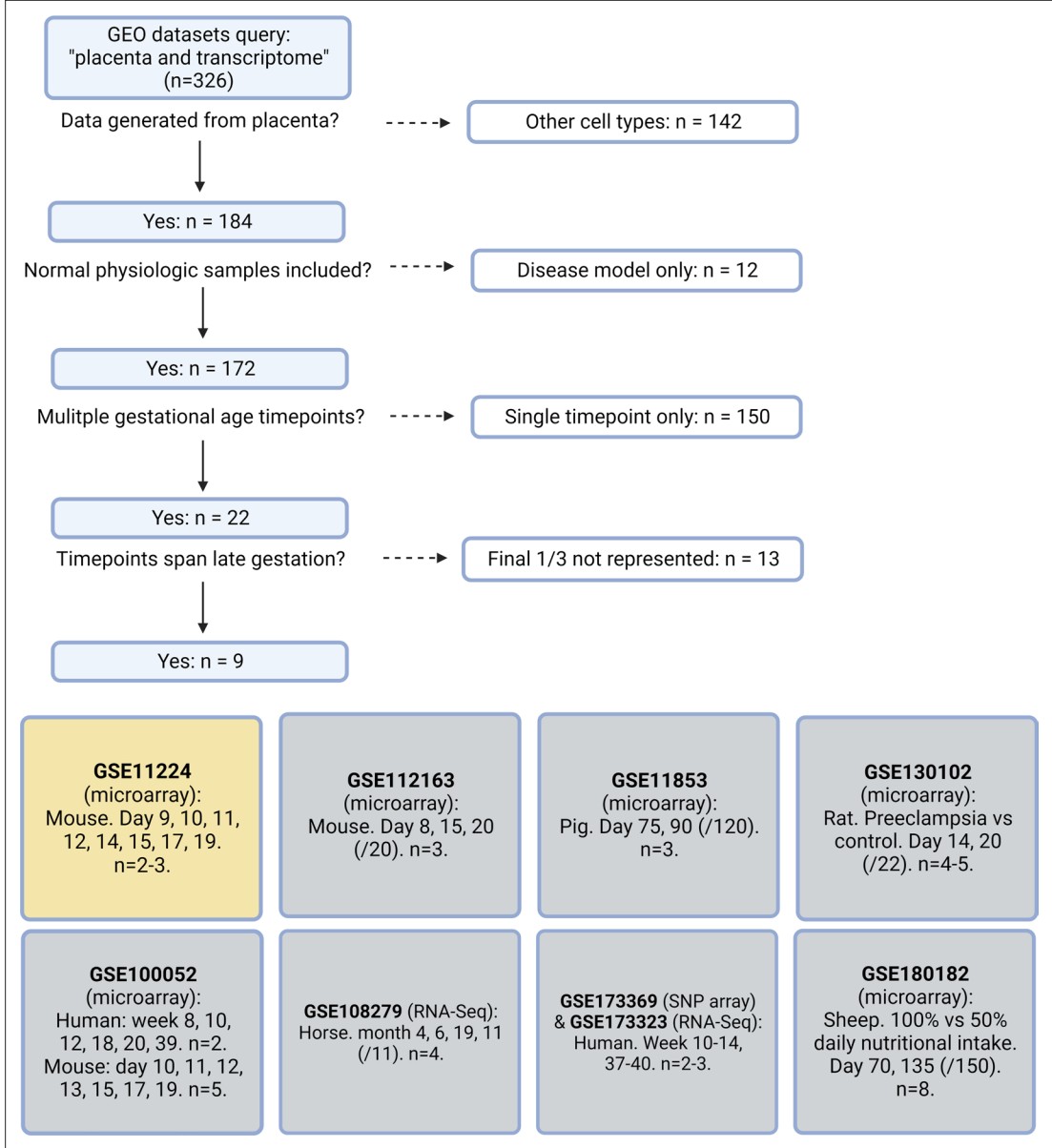

**Figure 1.** Systematic search flow for placental transcriptomics datasets. 326 Gene Expression Omnibus (GEO) datasets were identified by the search terms 'placenta' and 'transcriptome'; nine met criteria for containing placental transcriptomic data representing normal physiology at a range of gestational timepoints spanning through the final 1/3 of pregnancy. Dataset selected for further analysis in yellow.

among these genes include HIF-1 signaling, AMPK signaling, and cellular senescence. Genes in the 'turquoise' cluster showed increasingly negative correlation to advancing timepoints; KEGG functional pathways significantly overrepresented among these genes include the citric acid cycle, mitochondrial complex I biogenesis, and the mitotic cell cycle.

To validate the dynamic pathway activity suggested by the WGCNA, we used qPCR to quantify mRNA expression of the senescence marker *Glb1* (*Lee et al., 2006*) in our mouse placentas, which mirrored the WGCNA finding that cellular senescence in the placenta peaks in the final days of gestation (*Figure 2B*). HIF-1 protein abundance was found to peak at e17.5 (*Figure 2C*) and HIF-1 targets *Hk2* and *Slc2a1* likewise confirmed increasing HIF-1 activation with advancing gestational age through e17.5 (*Figure 2D*). Of note, a modest but significant fetal sex-dependent difference in HIF-1 target expression (but not *Glb1*) was also observed across timepoints (*Figure 2—figure supplement 1*). To assess changes in mitochondrial abundance across gestation, we measured COX IV protein expression and found it declined progressively across mouse gestation (*Figure 2E*) as predicted by the WGCNA.

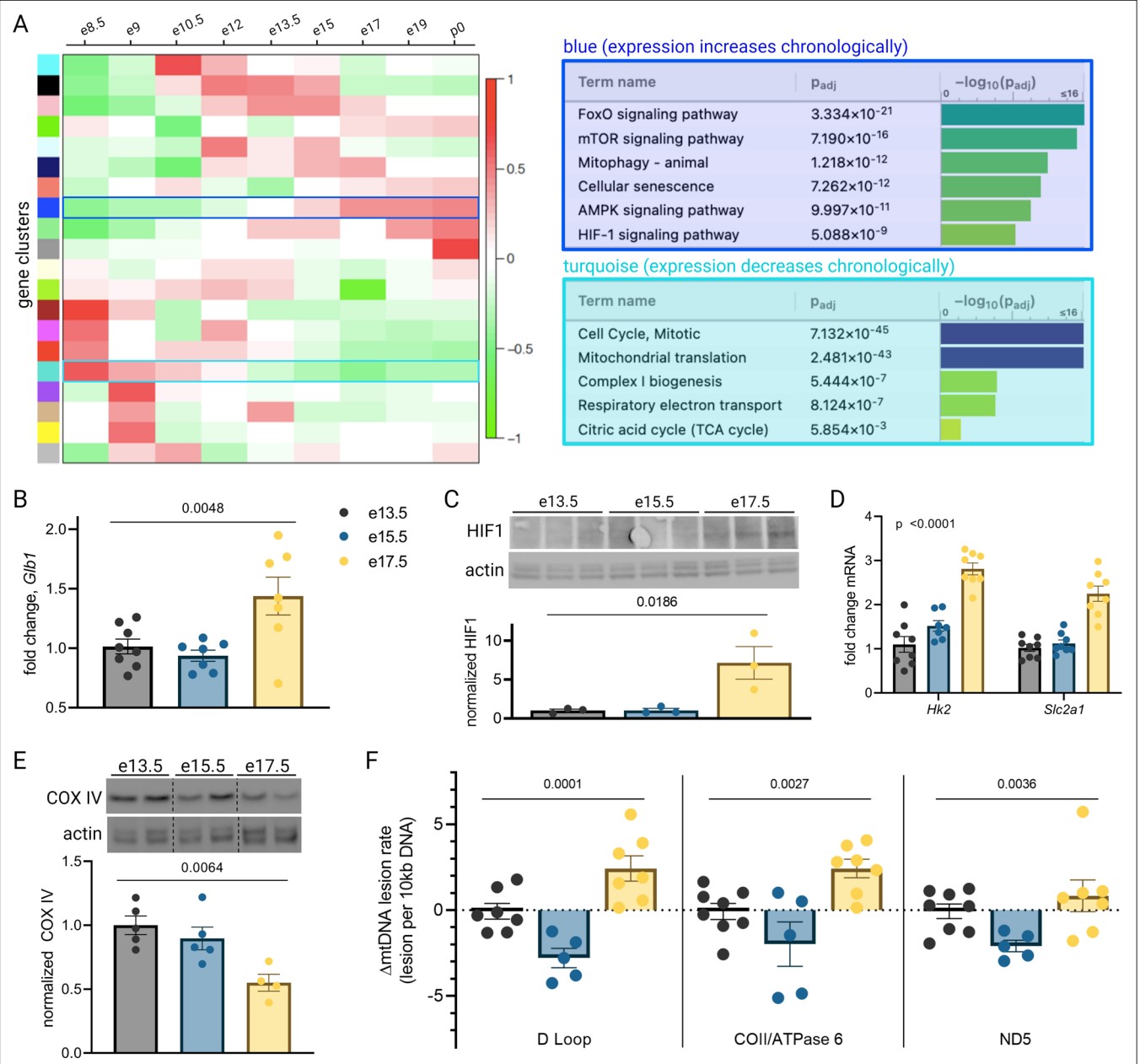

**Figure 2.** Mouse placental aging is characterized by cellular senescence, hypoxia-inducible factor 1 (HIF-1) signaling, and mitochondrial dysregulation. Weighted gene correlation network analysis (WGCNA) yielded 20 gene clusters. Functional pathways overrepresented in clusters found to increase (blue) and decrease (turquoise) across gestation highlight enhanced cellular senescence, increased HIF-1 signaling, and decreased mitochondrial synthesis and respiration late in pregnancy (**A**). mRNA expression of senescence marker *Glb1* peaks at e17.5 (B; one-way ANOVA p=0.0048). HIF-1 protein abundance is higher at e17.5 versus e13.5 and e15.5 (C; one-way ANOVA p=0.019), as is expression of HIF-1 targets *Hk2* and *Slc2a1* (D; two-way ANOVA p<0.0001 for gestational age factor). (See *Figure 2—figure supplement 1* for analysis of gene expression changes across timepoints by placental sex.) Mitochondrial abundance, reflected by COX IV protein, decreases with gestational age (E, one-way ANOVA p=0.0064), and mitochondrial DNA lesion rate peaks at e17.5 in the regions of the D-loop (one-way ANOVA p=0.0001), COII/ATPase6 (p=0.0027), and ND5 (p=0.036) (**F**). (B–F) Each data point represents a biological replicate (e.g. RNA, protein, or DNA extracted from an individual placenta, in turn collected from one of 2–4 pregnant dams per group). Data normalized to mean at e13.5. See *Figure 2—source data 1* for uncropped blots.

The online version of this article includes the following source data and figure supplement(s) for figure 2:

**Source data 1.** Uncropped, unedited blots from 2c (left) and 2e (right).

**Figure supplement 1.** Gestational age-dependent variability in expression of hypoxia-inducible factor 1 (Hif-1) target *Slc2a1*, but not *Hk2*, is affected by placental sex.

**Table 1.** Maternal and fetal characteristics.
Data summarized by mean ± SEM or *n* (%). p-Values calculated via t-test (continuous variables) or Chi-square contingency table (categorical variables).

| | <ins><35 weeks</ins> | <ins>>39 weeks</ins> | p-Value |
|---|---|---|---|
| | *n*=9 | *n*=11 | |
| Gestational age at delivery (weeks) | 34.0±0.3 | 39.5±0.1 | <0.0001 |
| Maternal age (years) | 31.9±0.9 | 36.2±1.0 | 0.008 |
| Maternal BMI >40 | 1 (11) | 0 (0) | 0.26 |
| Maternal race | | | |
| White | 7 (78) | 9 (82) | 0.13 |
| Black | 2 (22) | 0 (0) | |
| Asian | 0 (0) | 2 (18) | |
| Primiparous | 4 (44) | 4 (36) | 0.71 |
| Female neonate | 5 (55) | 8 (73) | 0.42 |
| Indication for delivery | Placenta previa (3) | Scheduled repeat (7) | |
| | Vasa previa (4) | Breech presentation (3) | |
| | Thinned lower uterine segment (2) | Elective (1) | |

Having observed a pattern of declining mitochondrial abundance in the placenta with advancing gestational age, we next investigated the mitochondrial DNA (mtDNA) lesion rate. The mitochondrial genome is particularly vulnerable to reactive oxygen species (ROS) insults, and mtDNA damage participates in a vicious cycle with mitochondrial dysfunction and further ROS production; these effects are observed in a number of age-related diseases in various tissues (*Jang et al., 2018*) and may drive age-associated loss of function (*Trifunovic et al., 2004*). We employed a semi-long run real-time qPCR approach (*Rothfuss et al., 2010*) to quantify relative mtDNA lesion rates in mouse placentas, normalized to e13.5. There was a measurable increase in the mtDNA lesion rate in the D-loop and COII/ATPase 6 regions of the mitochondrial genome at e17.5 (*Figure 2F*). Together, these results reflect a series of coordinated changes as gestation progresses—namely HIF-1 signaling induction, decreasing mitochondrial abundance, accumulating mtDNA damage, and escalating cellular senescence—confirming the patterns we discovered through reanalysis of published transcriptomic data.

We next probed human placentas for the same gestational age-dependent changes. Studying normal human placenta at a range of gestational ages beyond the second trimester is challenging, as placental sampling is usually only possible after delivery, and most deliveries associated with healthy pregnancies occur at term. We therefore sought to capitalize on rare exceptions such as cases of placenta previa, vasa previa, or uterine dehiscence, where iatrogenic preterm delivery is indicated for reasons unrelated to placental health. With an objective to compare term versus preterm placentas, yet minimize confounding factors that accompany labor and disease states that could be expected to affect placental health, we designed a case-control study using the following exclusion criteria: onset of labor prior to delivery (spontaneous or induced); maternal history of hypertension, asthma, diabetes, or autoimmune disease; pregnancy complicated by gestational hypertension, gestational diabetes, preeclampsia, multiples, fetal anomalies, placenta accreta spectrum disorder, or smoking during pregnancy.

Placentas from 9 cesarean deliveries occurring before 35 weeks' gestation and 11 cesarean deliveries occurring after 39 weeks' gestation were studied (*Table 1*). Maternal characteristics including race, nulliparity, and obesity were not statistically different across the early versus late groups; there was a small but statistically significant difference in maternal age at the time of delivery (31.9±0.9 years versus 36.2±1.0 for early gestation versus late, p=0.008). The distribution of fetal sex was not different among groups.

qPCR revealed a trend toward increased mRNA expression of *GLB1* and HIF-1 targets *HK2* and *SLC2A1* in the >39-week cohort (*Figure 3A*), consistent with the gestational age-dependent effect seen in mouse placenta. We also examined mitochondrial abundance in the two groups and found that mitochondrial RNA transcripts *ATP6* and *COX2* were significantly decreased (*Figure 3B*) and COX IV protein abundance was lower at the later timepoint (*Figure 3C*), mirroring the mouse findings. Of note, power calculation to reject the null hypothesis for difference between means for some of these measurements (with $\alpha$=0.05 and $\beta$=0.2) suggests a sample size of greater than 35 per group is required, assuming a similar effect size as seen in mouse data (e.g. expecting *GLB1* fold change [FC] difference of 40% across groups), and greater variability than seen for mouse data (e.g. expected standard deviation of *GLB1* FC equal to 0.6, vs 0.3 in mice). Practical constraints, especially given our strict exclusion criteria, make a study of this size unfeasible; nonetheless, we have included analysis of 20 human placentas here in recognition of the vital importance of translating mouse findings to human biology, even preliminarily. The data should be interpreted in the context of these statistical realities.

The co-appearance of cellular senescence, HIF-1 signaling, and mitochondrial dysregulation in the placenta as it approaches the end of gestation led us to hypothesize that in aging placental cells, HIF-1 induction could be upstream of mitochondrial dysregulation and cellular senescence, as is seen in other systems in emerging aging research (*Bratic and Larsson, 2013*; *Wiley and Campisi, 2016*; *Wiley and Campisi, 2021*). To test this hypothesis, we established a pharmacological model of HIF-1 induction in primary mouse trophoblasts using cobalt chloride, a prolyl hydroxylase inhibitor that stabilizes HIF-1$\alpha$ (*Maxwell et al., 1999*; *Jaakkola et al., 2001*) and has been widely used to model hypoxia. After 6 hr of CoCl$_2$ exposure, we confirmed HIF-1 protein accumulation in cultured trophoblasts (*Figure 4A*). After 48 hr of CoCl$_2$ exposure, mouse trophoblasts exhibit decreased mitochondrial abundance, by *Cox2* mRNA expression (*Figure 4B*) and Cox IV protein abundance (*Figure 4C*), and an increase in senescence-associated beta galactosidase (SA-$\beta$Gal), encoded by *Glb1* (*Figure 4D*) and detected as a blue stain in an X-gal assay for senescence (*Figure 4E*). These findings suggest that HIF-1 stabilization induces subsequent mitochondrial dysfunction and senescence in trophoblasts.

Primary trophoblasts undergo spontaneous syncytialization in culture, a phenomenon that limits the duration of study and may also confound the interpretation of experimental changes in key metabolic factors (*Nursalim et al., 2020*). We therefore also modeled HIF-1 activation in JAR cells, a trophoblast cell line that does not undergo syncytialization (*Rothbauer et al., 2017*). Consistent with our results in primary cells, we found that HIF-1 is stabilized in CoCl$_2$-treated JAR cells (*Figure 5A*), and after 6 days of exposure, mtDNA and protein abundance declined (*Figure 5B–C*). We further defined the time course of mitochondrial downregulation: by 72 hr the effect began to appear (*Figure 5— figure supplement 1*). Furthermore, at the 6-day timepoint we found that CoCl$_2$ exposure leads to accumulation of mitochondrial ROS, as measured by mtSOX, a fluorescent mt-superoxide indicator dye (*Figure 5D*), and impairs mitochondrial polarization, as measured by tetramethylrhodamine ethyl ester (TMRE) staining, a cell-permeant fluorescent dye that accumulates in polarized mitochondria (*Figure 5E*). Additionally, we observed pronounced signs of cellular senescence: morphological hallmarks (cellular swelling) and $\beta$-galactosidase overexpression (*Figure 5F*); growth arrest which persists for days after removal of the HIF-1 stabilizing compound (*Figure 5G*); and a senescence-associated secretory phenotype (SASP, *Figure 5H*) reflected by increases in mRNA expression of the genes encoding VEGF, TNF$\alpha$, and IL-1$\alpha$ and a decrease in mRNA expression of the gene encoding anti-inflammatory cytokine IL-10.

We conducted additional studies to confirm that cell death was not a primary contributor to the lack of proliferation. Propidium iodide staining with quantitative fluorescence cytometry indicated that CoCl$_2$ treatment only increased cell death by 0–3% (*Figure 5—figure supplement 2*). Importantly, cells remained adherent and continued to acidify culture medium beyond 14 days of CoCl$_2$ exposure, providing confidence that HIF-1 stabilization induces a phenotype characterized by predominantly viable cells that are no longer proliferating, namely cellular senescence. Finally, to assess whether the effects we observed were attributable to HIF-1 stabilization and not an off-target effect of CoCl$_2$, we also evaluated an alternative prolyl hydroxylase inhibitor, dimethyloxalylglycine (DMOG) (*Epstein et al., 2001*), and found similar effects on HIF-1, mitochondrial abundance and cellular senescence (*Figure 5—figure supplement 3*).

To determine if other features of the aged placenta phenotype were recapitulated in this model, we performed whole transcriptome analysis via RNA-Seq. Differential expression (log2 FC>1;

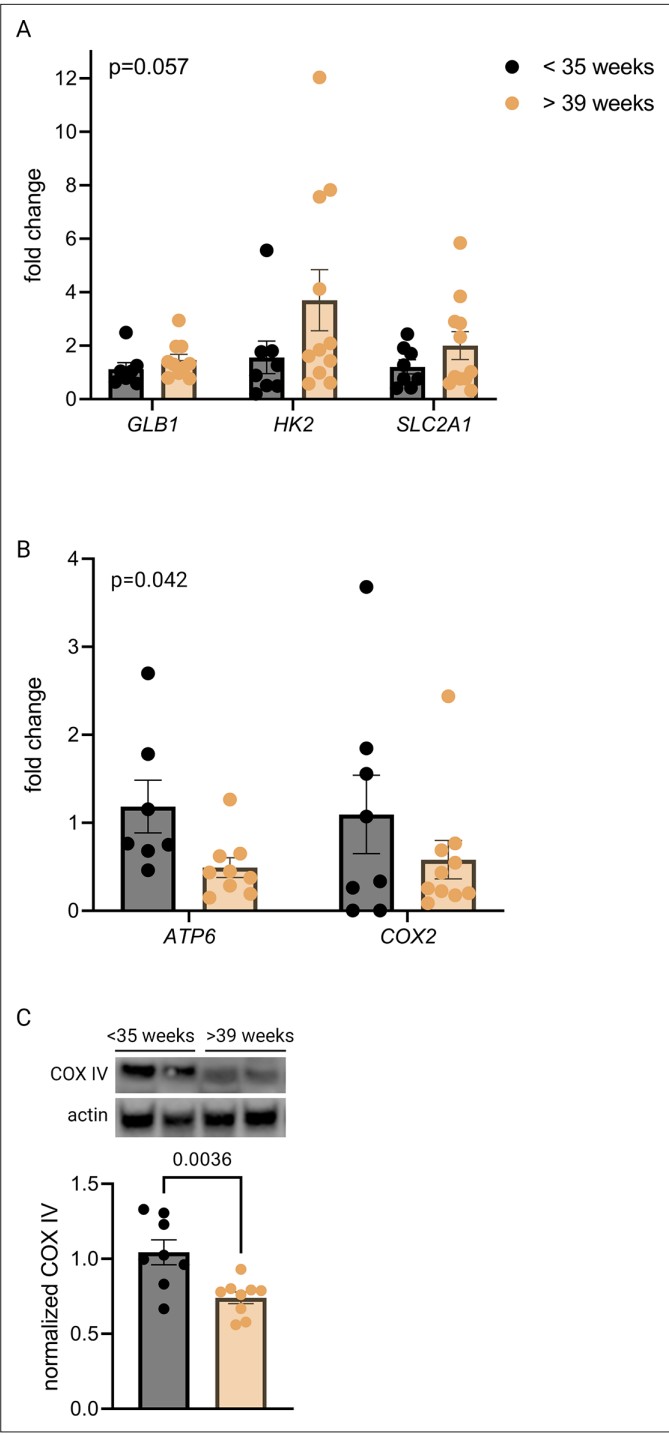

**Figure 3.** Senescence, hypoxia-inducible factor 1 (HIF-1) signaling, and decreased mitochondrial abundance characterize late-gestation human placenta. mRNA expression of senescence marker *GLB1* and HIF-1 targets *HK2* and *SLC2A1* trends higher in placentas from >39-week cohort vs <35-week cohort (A; two-way ANOVA gestational age factor p=0.057). Mitochondrial abundance, reflected by mitochondrial genes *ATP6* and *COX2* (B; two-way ANOVA gestational age factor p=0.042) and COX IV protein (C; p=0.0036) decreases with advancing gestational age. Each data point represents a biological replicate (RNA or protein isolated from an individual placenta). Data normalized to mean in <35-week group. See *Figure 3—source data 1* for uncropped blots.

The online version of this article includes the following source data for figure 3:

**Source data 1.** Uncropped, unedited blots from *Figure 3*.

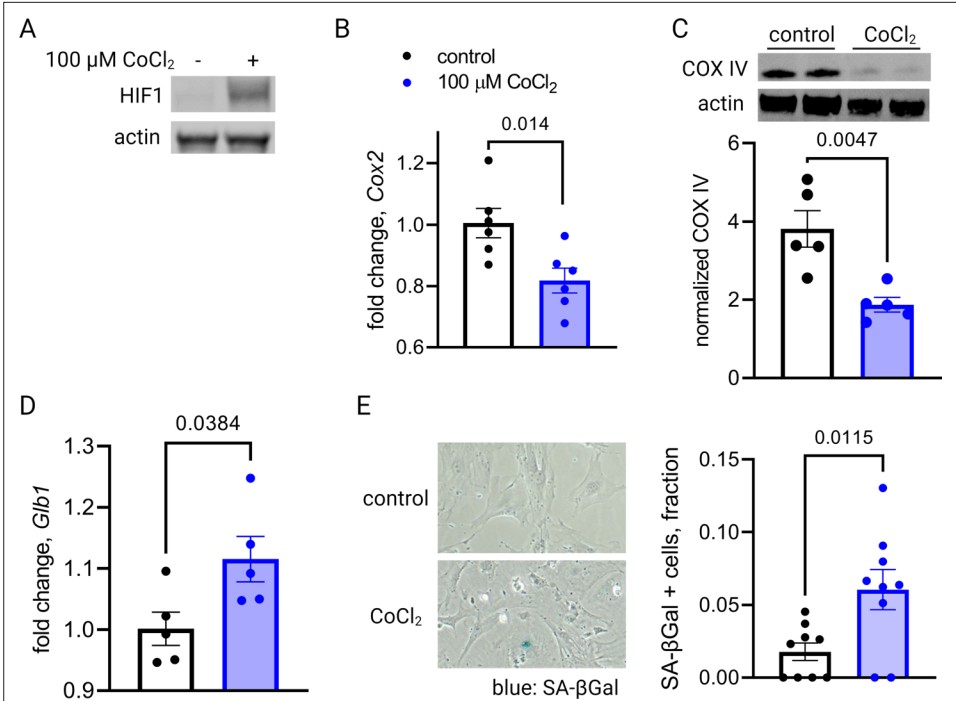

**Figure 4.** Short-term hypoxia-inducible factor 1 (HIF-1) stabilization in primary mouse trophoblasts leads to decreased mitochondrial abundance and cellular senescence. HIF-1 is detected in cultured trophoblasts exposed to CoCl₂ (**A**). After 48 hr of CoCl₂ exposure, trophoblasts exhibit decreased mitochondrial abundance reflected by *Cox2* mRNA expression levels (B; p=0.014) and COX IV protein levels (C; p=0.0047). Senescence marker *Glb1* is increased (D; p=0.038) and senescence-associated beta galactosidase (SA-βGal) accumulation is noted by X-gal assay (E; p=0.012). Each data point represents a technical replicate (e.g. protein, RNA, or β-Gal measured from an individual well of cells grown in treated or control condition). Data normalized to mean of control treatment group. See *Figure 4—source data 1* for uncropped blots.

The online version of this article includes the following source data for figure 4:

**Source data 1.** Uncropped, unedited blots from 4a (left) and 4c (right).

false discovery rate [FDR] <0.05) was found for 2188 upregulated and 1389 downregulated genes (*Figure 5I*). Gene set enrichment analysis revealed that chemical hypoxia led to upregulation of inflammatory signaling, and downregulation of the TCA cycle, mitochondrial biogenesis, and respiratory electron transport. These data therefore recapitulated metabolic patterns that emerged from time course transcriptomic analysis of intact placentas (*Figure 2A*).

The onset of labor is accompanied by a transformation of the uterus from quiescence into a distinct physiologic state in which it generates powerful, coordinated contractions. This phenotypic change is characterized by upregulation of a cadre of contraction-associated proteins (CAPs), notably cyclo-oxygenase-2 (COX-2, encoded by *PTGS2*), prostaglandin F2α receptor (encoded by *PTGFR*), interleukin-6 (IL-6), and connexin 43 (Cx43, encoded by *GJA1*). Phenotypic switching can be modeled in primary and immortalized uterine myocytes upon stimulation with inflammatory mediators such as IL-1β, PGF2α, and thrombin (*Nishimura et al., 2020*; *Rauk and Chiao, 2000*; *Leimert et al., 2019*).

To test whether primary placental metabolic disruptions could crosstalk with uterine myocytes to trigger the contractile phenotype, we collected conditioned media from our JAR cell model of HIF-1-driven senescence and measured the effect on expression of CAPs in uterine myocytes (hTERT-HM cells). We observed a robust and specific effect: co-stimulation of uterine myocytes with IL-1β plus conditioned media from JAR cells treated with CoCl₂ (but not from untreated JAR cells) potentiated the induction of *PTGS2*, *GJA1*, *PTGFR*, and *IL6* mRNA expression (*Figure 6A–D*). We next employed a collagen lattice assay previously described for assessing contractility of myometrial cells in vitro (*Nishimura et al., 2020*; *Devost and Zingg, 2007*), and demonstrated that myocyte contraction is augmented upon exposure to conditioned media from JAR cells treated with CoCl₂ but not from

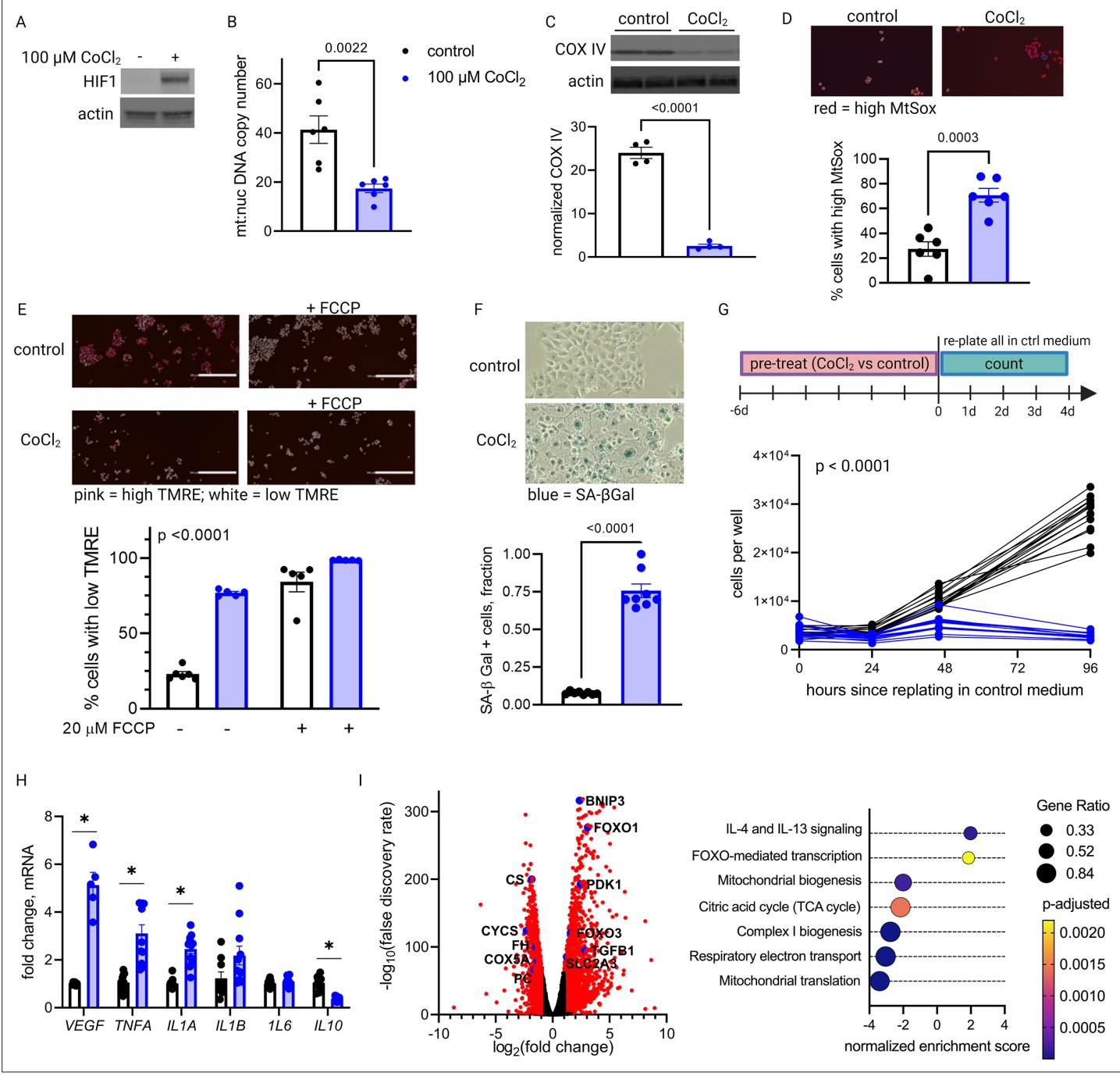

**Figure 5.** Long-term hypoxia-inducible factor 1 (HIF-1) stabilization in JAR cells leads to mitochondrial dysfunction, cellular senescence, and metabolic reprogramming. HIF-1 is stabilized at 6-day timepoint of CoCl$_2$ exposure (**A**). After 6 days, mitochondrial abundance is decreased as reflected by a drop in the mitochondrial:nuclear DNA copy number (**B**) and a decrease in COX IV protein (**C**). (See ***Figure 5—figure supplement 1*** for timecourse of declining mitochondrial abundance.) Cells also exhibit augmented signs of mitochondrial dysfunction via MtSox (**D**; p=0.0003) and tetramethylrhodamine ethyl ester (TMRE) staining (**E**; two-way ANOVA CoCl$_2$ factor p<0.0001). Senescence-associated beta galactosidase (SA-βGal) staining reflects a high proportion of senescent cells (**F**; p<0.0001) and growth arrest is confirmed by cell counting following a 6-day pre-treatment with CoCl$_2$ (**G**; two-way ANOVA p<0.0001 for interaction of CoCl$_2$ factor with time). (See ***Figure 5—figure supplement 2*** for assessment of cell death by propidium iodide staining.) mRNA expression of senescence-associated secretory phenotype (SASP) candidates *VEGF, TNFA, IL1A*, and *IL10* is altered after CoCl$_2$ exposure (**H**; *, adjusted p<0.01). RNA-Seq revealed upregulation of 2188 and downregulation of 1389 genes (**I**; genes with |log2(FC)|>1 and -log(FDR)>2 indicated in red) after CoCl$_2$ treatment, with gene set enrichment analysis revealing several pathways significantly dysregulated after CoCl$_2$ treatment recapitulating changes seen in transcriptomic analysis of late versus early gestation mouse placenta. Scale marker = 200 μm. FCCP = carbonyl cyanide 4-(trifluoromethoxy) phenylhydrazone, an ionophore uncoupler of oxidative phosphorylation which depolarizes mt membrane potential. See

*Figure 5 continued on next page*

Figure 5 continued

*Figure 5—figure supplement 3* for assessment of effects of HIF-1 stabilization in JAR cells using dimethyloxalyl glycine (DMOG). Each data point represents a technical replicate (measurement from an independent well of cells grown in treatment vs control condition). Data normalized to mean of control group. See *Figure 5—source data 1* for uncropped blots.

The online version of this article includes the following source data and figure supplement(s) for figure 5:

**Source data 1.** Uncropped, unedited blots from 5a (left) and 5c (right).

**Figure supplement 1.** The mitochondrial effects of hypoxia-inducible factor 1 (HIF-1) stabilization in JAR cells begin to appear on day 3 following $CoCl_2$ exposure.

**Figure supplement 2.** Increased number of JAR cells stain with propidium iodide, but the absolute number remains low following 6 days of $CoCl_2$ treatment.

**Figure supplement 3.** Dimethyloxalyl glycine (DMOG) stabilizes hypoxia-inducible factor 1 (HIF-1) in JAR cells (**A**) and induces similar effects as $CoCl_2$ on COX IV protein (**B**), senescence-associated beta galactosidase (SA-βGal) expression (**C**), and cell growth (**D**) after 4 days.

untreated JAR cells (*Figure 6E*). These transcriptional and functional results collectively demonstrate that uterine myocytes are responsive to the secretome of JAR cells driven to senescence via HIF-1 induction.

To test whether HIF-1 induction drives labor onset in vivo, we administered DMOG intraperitoneally to pregnant mice on gestational day e16.5. Analysis of placentas recovered 12 hr following injection indicated that HIF-1 protein is stabilized in placenta following maternal DMOG injection (*Figure 7A*), and transcription of HIF-1 target genes *Hk2* and *Slc2a1* was significantly increased (*Figure 7B*). Following injection of DMOG, gestational length was significantly shortened compared to injection of vehicle alone (*Figure 7C–D*).

## Discussion

The subject of whether the healthy placenta undergoes aging (with accompanying dysfunction) within its 40-week lifespan has been of great interest and debate for many years. Age-related functional decline is a near-universal phenomenon affecting many tissues, but the underlying biochemical driving factors vary widely depending on the context of cell type and local stressors. Mechanisms well described in other tissues are rooted in genomic instability, mitochondrial dysfunction and oxidative stress, nutrient deprivation and metabolic insufficiency, or loss of proteostasis (*Campisi et al., 2019*). While characterization of the placenta as an aging organ appeared in the literature beginning nearly 50 years ago (*Rosso, 1976*; *Martin and Spicer, 1973*), a counter-argument has been offered that there is no logical reason for the placenta to undergo accelerated aging relative to the fetus, since both share the same genes and environment (*Fox, 1997*). A third viewpoint proposes that aging in the placenta is indicative of a disease state—not normal progression of healthy gestation—one usually reflective of maladaptive responses to oxidative stress (*Sultana et al., 2017*; *Cindrova-Davies et al., 2018*; *Biron-Shental et al., 2010*; *Chen et al., 2011*; *Maiti et al., 2017*). Finally, recent reports have noted prominent signs of aging in another fetally derived tissue, the chorioamniotic membranes, in healthy pregnancies (*Bonney et al., 2016*), and have linked aging of the fetal membranes to the onset of labor (*Menon et al., 2016*; *Menon et al., 2019*). The results presented here span mouse and human placentas to provide evidence for placental aging in normal pregnancy. Mechanistic dissection of this phenomenon further demonstrates how hypoxia in late gestation may be both a causal determinant of mitochondrial dysfunction and senescence observed within the trophoblast as well as a trigger to induce uterine contraction.

To our knowledge, this is the first report to propose that placental aging in healthy pregnancy is characterized by induction of HIF-1 signaling, accompanied by downstream effects including mitochondrial dysfunction and cellular senescence. These findings have implications for the role of the placenta in signaling that promotes the onset of labor, particularly given our demonstration that upon HIF-1 stabilization, trophoblasts can induce inflammatory changes in uterine myocytes and potentiate their contractility, and our in vivo demonstration that administration of DMOG stabilizes placental HIF-1 and leads to preterm labor in mice. Important next steps prompted by this work include determining the specific components of the secretome of HIF-1-activated trophoblasts which are responsible for inducing myocyte transformation, anticipating overlap with findings of prior studies

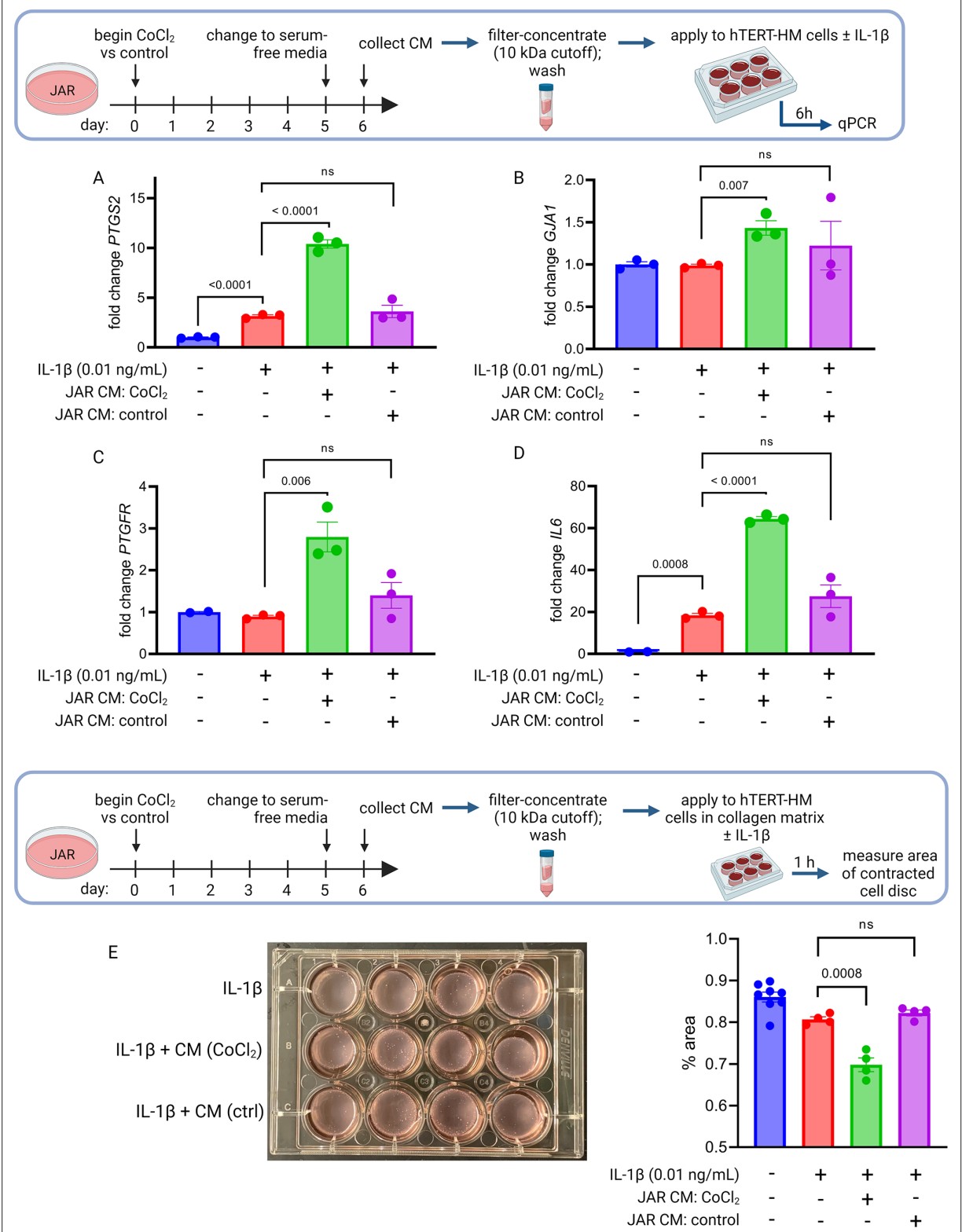

**Figure 6.** Conditioned media (CM) from JAR cells following hypoxia-inducible factor 1 (HIF-1) stabilization induces expression of contractile-associated proteins and augments contraction in immortalized human uterine myocytes. hTERT-HM mRNA expression of *PTGS2* (**A**), *GJA1* (**B**), *PTGFR* (**C**), and *IL6* (**D**) was induced by CM from JAR cells following $CoCl_2$ treatment (but not in control conditions), potentiating the effect of stimulation of myocytes by exogenous IL-1β. Data normalized to mean of null treatment group. Percent well area occupied by hTERT-HM cells embedded in collagen matrix is significantly smaller after stimulation with IL-1β plus JAR CM from $CoCl_2$ condition, reflecting greater degree of hTERT-HM cellular contraction (**E**). Each data point represents a technical replicate (measurement from an independent well of cells grown in treatment vs control condition).

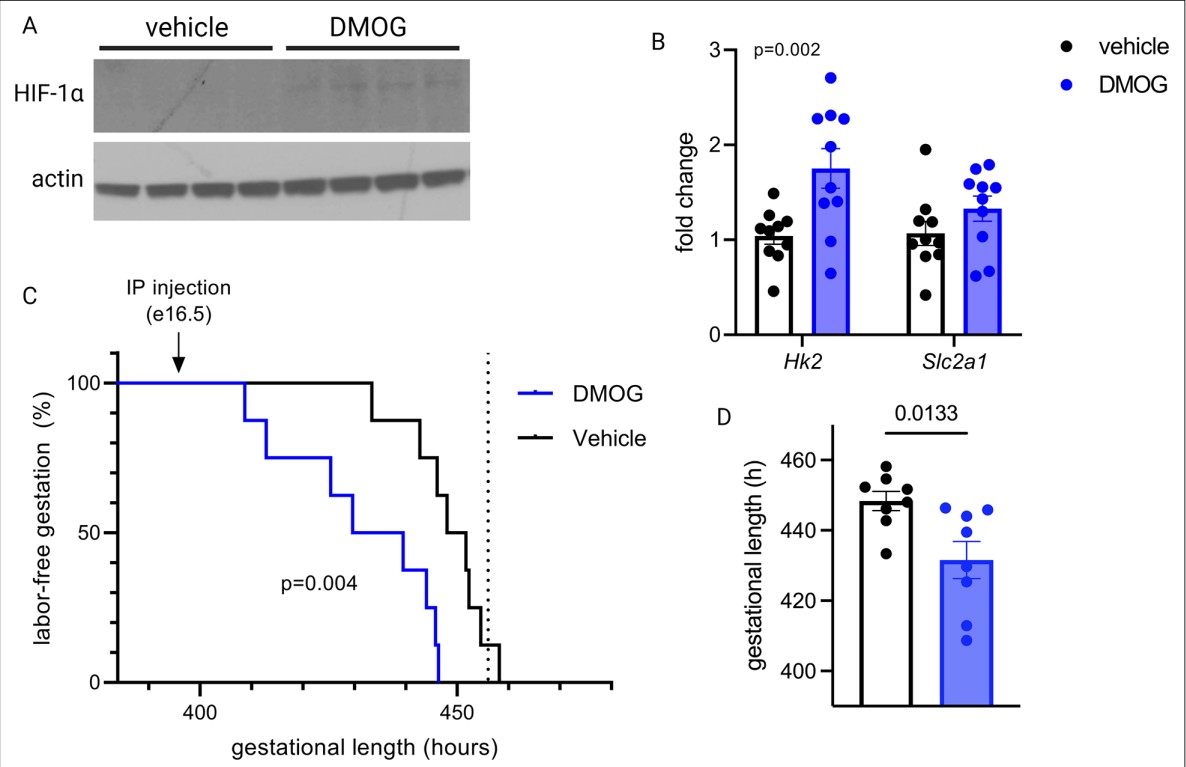

**Figure 7.** Maternal dimethyloxalyl glycine (DMOG) injection on e16.5 stabilizes placental hypoxia-inducible factor 1 (HIF-1) and induces preterm labor. HIF-1α protein is detected in placental lysates 12 hr following DMOG injection but not vehicle (**A**). mRNA expression of HIF-1 targets *Hk2* and *Slc2a1* is upregulated following DMOG injection (p=0.002 for DMOG vs vehicle, two-way ANOVA) (**B**). Gestational length is significantly shortened following DMOG injection versus vehicle (**C–D**). Each data point represents a biological replicate (in A and B, each measurement from an individual placenta collected from one of two pregnant dams). Data normalized to mean of vehicle group. See *Figure 7—source data 1* for uncropped blots.

The online version of this article includes the following source data for figure 7:

**Source data 1.** Uncropped, unedited blots from *Figure 7*.

delineating paracrine signals that mediate labor onset (*Sheller-Miller et al., 2019*; *Migale et al., 2015*; *Srikhajon et al., 2014*; *Gomez-Lopez et al., 2014*). Additionally, having established that HIF-1 signaling is upstream of mitochondrial dysfunction and cellular senescence in late-gestation placenta in normal pregnancies, future studies can investigate whether factors that modulate this pathway could increase or decrease vulnerability to preterm labor. In studies targeted to uterine decidua, a link between cellular senescence and labor onset has previously been established, with evidence implicating phospho-Akt and mTORC1 signaling upstream of prostaglandin synthesis as a key mechanism in this model (*Hirota et al., 2010*; *Hirota et al., 2011*). It is possible that senescence signaling from the placenta and decidua converge, and labor is provoked upon these convergent endpoint signals surpassing a threshold, regardless of their origin within the gestational compartment. Future studies using targeted manipulation of senescence and metabolism in these compartments individually will help clarify their specific contributions to labor onset.

Our results suggest that placental HIF-1 becomes stabilized late in gestation, and this could be secondary to the onset of hypoxia sensing as oxygen demand outstrips supply in the fetoplacental unit. However, placental hypoxia has been difficult to examine in vivo. Despite evolving approaches for measurement of placental oxygenation, findings have varied widely and adequate spatiotemporal resolution has so far been challenging (*Nye et al., 2018*). However, it is apparent that due to arterio-venous shunting and high metabolic extraction rates, the placenta experiences hypoxia throughout pregnancy with a reported partial pressure of oxygen 30–50 mm Hg in the intervillous space, so perhaps a relevant question is what *prevents* HIF-1 stabilization and its downstream effects early in gestation. It is possible that the gestational age-dependent HIF-1 induction we observed reflects not a change in oxygen availability, but rather a change in other HIF-regulatory factors such as nicotinamide

adenine dinucleotide (NAD⁺) depletion, as has been demonstrated to occur in muscle and is accompanied by metabolic reprogramming and mitochondrial decline (*Gomes et al., 2013*). Importantly, replenishment of NAD⁺ was shown to restore markers of mitochondrial function in these aged cells, suggesting that mitochondrial effects of HIF-1-driven aging are reversible in some systems. Separately, a recent study which compared the miR-nomes of first- versus third-trimester human placenta highlighted differential expression of miRNAs with functional links to silencing of hypoxia response and cellular senescence pathways in the first trimester (*Gonzalez et al., 2021*). Future investigations into the impact of placental HIF-1 regulation will require a more precise understanding of placental oxygenation in vivo and a sophisticated approach to model these factors in placental explants or other cellular systems.

The shift in metabolic phenotype away from oxidative respiration in late pregnancy as demonstrated by our transcriptomics analysis is consistent with earlier findings from respirometry studies in mouse placenta (*Sferruzzi-Perri et al., 2019*), where rates of oxidative phosphorylation were noted to decline from e14 to e19, particularly in the labyrinthine zone. It is still unclear if this shift reflects changing substrate availability, or oxygen content, or perhaps the metabolic needs of the developing fetus. A better understanding of the dynamic metabolism of the placenta across gestation will be crucial for developing strategies for optimizing placental health, with major implications for both maternal and fetal outcomes. For example, placental metabolomics studies with a gestational time series could help corroborate the metabolic implications interpreted from our transcriptomic data. And while our study stemmed from analysis of the whole-organ placental transcriptome, single-cell transcriptomics across advancing gestational age in healthy placentas would offer critical cell-type localization of specific metabolic events noted to occur late in pregnancy.

Our data have implications for other adverse pregnancy outcomes beyond preterm birth, including intrauterine growth restriction and preeclampsia. Two important recent studies have examined separate mouse models with constitutively active placental HIF-1 and found that when induced from the beginning of gestation, placental HIF-1 signaling drives abnormal placentogenesis and impaired spiral artery remodeling, leading to placental hypoperfusion, fetal growth restriction, and a preeclampsia-like clinical syndrome (*Albers et al., 2019*; *Sallais et al., 2022*). In the context of these earlier results, our study highlights the critical gestational age dependence of HIF-1 effects, having demonstrated that in a normally developed mouse placenta, induction of HIF-1 signaling during the final days of gestation leads to labor onset. Timing may be key to understanding how two distinct pregnancy complications—preterm labor and preeclampsia—appear to emerge separately from overlapping pathophysiologic mechanisms (*Rasmussen et al., 2017*; *Mandò et al., 2014*; *Fujimaki et al., 2011*; *Davy et al., 2009*; *Burton et al., 2009*).

In summary, we report a molecular characterization of placental aging phenomena occurring in normal pregnancies, stemming from induction of HIF-1 signaling and downstream mitochondrial dysfunction and cellular senescence. These findings establish aging in the placenta as a feature of normal gestation; identify HIF-1 signaling as an upstream trigger leading to mitochondrial dysfunction and senescence in the placenta; demonstrate that the secretome of senescent trophoblasts is sufficient to potentiate uterine myocyte transformation and contractility; and establish that HIF-1 induction in vivo can induce preterm labor. These findings may have important implications for illuminating the factors that determine gestational length both in health and disease.

## Materials and methods

### Mouse placenta collection

Wild-type C57BL/6 mice (Jackson Laboratory #000664) were fed a chow diet and housed at 20°C in a 12 hr light/12 hr dark cycle. Nulliparous females (<6 months of age) were housed for a single dark cycle (midnight = gestational day 0) with a stud male. On gestational day e13.5–17.5, placentas were isolated via laparotomy from pregnant females anesthetized via surgical-plane isoflurane. Placentas were quartered, immersion-rinsed in dH₂O and blotted dry, then snap-frozen in liquid nitrogen and stored at –80°C prior to use.

### Mouse trophoblast isolation

Protocol described in full in *Pennington et al., 2012*. On gestational day e15.5, placentas were collected via laparotomy from pregnant wild-type C57BL/6 female mice anesthetized with

surgical-plane isoflurane. After dissection, placentas were quartered, immersion-rinsed in dH$_2$O, then placed in ice-cold digestion buffer: DMEM plus HEPES 20 mM, collagenase 1 mg/mL, and DNase I 4000 U/mL. Placentas were digested for 20–40 min at 37°C with trituration and examined under a microscope until optimal digestion achieved. Digested cells were passed through a 70 µm strainer, washed in DMEM, then separated using a Percoll gradient. Trophoblast fraction was washed once more then plated in complete medium: DMEM with HEPES 20 mM, FBS 10%, and supplemented with sodium pyruvate, pen-strep-glutamine, non-essential amino acids, and gentamicin. Cells were grown at 37°C in 95% ambient air with 5% CO$_2$.

## Mouse DMOG injection and measurement of gestational length

On gestational day e16.5, dams received an intraperitoneal injection of DMOG (Selleck Chemicals) 7.5 mg in 0.3 mL sterile saline (approximately 250 mg/kg) versus vehicle alone. Video recordings were used to measure gestational length, defined as interval from midnight of timed mating period until birth of the first pup.

## Human placental specimens

Placenta samples were collected at the time of cesarean delivery, within 20 min of delivery of the placenta. Placental tissue was sampled from approximately 1 cm deep to the maternal surface after dissecting away membranes. Cotyledons from all four quadrants were collected and minced together prior to immersion rinse in dH$_2$O and storage of separate aliquots by snap freezing versus immersion in RNA later. All samples were stored at –80°C prior to use.

## Cell culture

All cells were grown at 37°C in 95% ambient air with 5% CO$_2$. JAR choriocarcinoma cells (ATCC HTB-144; authenticated by STR profiling and confirmed negative for mycoplasma contamination) were grown in RPMI 1640 media (4.5 g/L glucose) supplemented with 10% FBS. Adherent JAR cells were treated with HIF-1-stabilizing agents dissolved in 1X PBS (pH 7.4): cobalt chloride hexahydrate (100 µM CoCl$_2$, Millipore Sigma) and dimethyloxalylglycine, *N*-(methoxyoxoacetyl)-glycine methyl ester (1 mM DMOG, Millipore Sigma). JAR cells were treated with 100 µM CoCl$_2$ or 1 mM DMOG in culture media as indicated prior to endpoint assays.

### Preparation of conditioned media

On day 5 of JAR cell treatment with CoCl$_2$ (versus control condition), media exchange was performed to apply serum-free media for both conditions. 24 hr later, conditioned media samples were collected and applied to 10 kDa molecular weight cutoff filters (Amicon Ultra), with centrifugation for 20 min at 4000 × *g*, 4°C. Filters were washed with one volume of hTERT-HM base medium (see below), with repeat centrifugation. Concentrated proteins retained in suspension above the filter were collected and stored at –80°C.

hTERT-HM cells (generously supplied by Dr. Jennifer Condon, Wayne State; STR reference profile not available) were grown in DMEM/F12 medium (Gibco) with 10% FBS and Antibiotic-Antimycotic (1X, Gibco). Cells were stimulated by addition of recombinant human IL-1β (R&D Systems) and/or filter-concentrated JAR cell conditioned media as indicated, 6 hr prior to collection of cellular RNA.

### RT-qPCR

Total RNA was reverse-transcribed to cDNA with the Superscript III reverse transcriptase (Invitrogen) system with random hexamer primers (Invitrogen) according to the manufacturer's instructions. cDNA was amplified via real-time quantitative PCR with TaqMan Fast Advanced Master Mix (Applied Biosystems) or SYBR Green PCR Master mix (QIAGEN) in a QuantStudio 6 Flex Real-Time PCR System (Applied Biosystems). In mouse samples, target gene expression was normalized to endogenous levels of housekeeping gene, β-actin. In human samples, target gene expression was normalized to endogenous levels of housekeeping gene, YWHAZ (TaqMan) or β-actin (SYBR). Mouse and human primers are listed in *Table 2*.

**Table 2.** Primers.

**Mouse**

| Target | Sequence | Platform |
|---|---|---|
| Glb1 | Mm00515342_m1 | TaqMan |
| Hk2 | Mm00443385_m1 | TaqMan |
| Slc2a1 | Mm00441473_m1 | TaqMan |
| D loop (long, 801 bp amplicon) | F: CGTACATTAAACTATTTTCCCCAAG<br>R: GAGTTTTGGTTCACGGAACAT | SYBR |
| COII/ATPase 6 (long, 855 bp amplicon) | F: TTGGTCTACAAGACGCCACA<br>R: ATTTTGGTGAAGGTGCCAGT | SYBR |
| Nd5 (long, 930 bp amplicon) | F: CGCCTACTCCTCAGTTAGCC<br>R: ATGGTGACTCAGTGCCAGGT | SYBR |
| Nd2/Nd1 (long, 832 bp amplicon) | F: GGATGAGCCTCAAACTCCAA<br>R: ATGATGGCAAGGGTGATAGG | SYBR |
| D loop (short) | F: TGACTATCCCCTTCCCCATT<br>R: TTGTTGGTTTCACGGAGGAT | SYBR |
| COII/ATPase 6 (short) | F: TCTCCCCTCTCTACGCATTC<br>R: CGGTTAATACGGGGTTGTTG | SYBR |
| Nd5 (short) | F: GGCCTCACATCATCACTCCT<br>R: GCTGTGGATCCGTTCGTAGT | SYBR |
| Nd2/Nd1 (short) | F: GGATGAGCCTCAAACTCCAA<br>R: GGCTCGTAAAGCTCCGAATA | SYBR |
| Actb | Mm00607939_s1 | TaqMan |

**Human**

| Target | Sequence | Platform |
|---|---|---|
| HK2 | Hs00606086_m1 | TaqMan |
| SLC2A1 | Hs00892681_m1 | TaqMan |
| IL1A | Hs00174092_m1 | TaqMan |
| IL1B | Hs01555410_m1 | TaqMan |
| TNFA | Hs00174128_m1 | TaqMan |
| IL6 | Hs00174131_m1 | TaqMan |
| IL10 | Hs00961622_m1 | TaqMan |
| MT-ATP6 | Hs02596862_g1 | TaqMan |
| MT-COX2 | Hs02596865_g1 | TaqMan |
| ND1 | F: CCATAAAACCCGCCACACT<br>R: GAGCGATGGTGAGAGCTAAGGT | SYBR |
| 18S | F: CGCAGCTAGGAATAATGGAATAGG<br>R: CATGGCCTCAGTTCCGAAA | SYBR |
| GJA1 | Hs.PT.58.38338544 | SYBR |
| PTGS2 | Hs.PT.58.77266 | SYBR |
| ACTB | Hs.PT.39a.22214847 | SYBR |
| YWHAZ | Hs01122445_g1 | TaqMan |

## Collagen lattice contraction assay

As previously described (**Nishimura et al., 2020**; **Devost and Zingg, 2007**), hTERT-HM cells were suspended in a collagen gel matrix (Cellmatrix collagen type I-A, Fisher Scientific, prepared with MEM, NaOH, and HEPES) then plated in 12-well dishes and grown in hTERT-HM medium. When cells reached approximately 60% confluence, gel matrix was detached from plate and experimental

**Table 3.** Antibodies.

| Antibody | Species | Working concentration | Source |
|---|---|---|---|
| CoxIV | Mouse IgG mAb | 1:1000 | Cell Signaling Technology, #11967S |
| β-Actin-HRP conjugate | Rabbit IgG mAb | 1:2000 | Cell Signaling Technology, # 5125S |
| HIF-1 | Rabbit | 1:1000 | Cell Signaling Technology, #14179S |

treatments were applied in fresh cell culture medium. After 1 hr incubation, a photograph was taken of each well for manual measurement of percent well area occupied by collagen disc (using ImageJ), reflecting cellular contraction.

## mtDNA damage assay

mtDNA damage was assayed from total DNA via a semi-long run real-time PCR approach, as described elsewhere (*Rothfuss et al., 2010*). Briefly, separate qPCRs were assayed to compare a long versus short amplicon representing each genomic region. A lesion rate per 10 kb of mtDNA was calculated as:

$$\text{lesion rate } = \frac{10000 \ (bp)}{size \ of \ long \ fragment \ (bp)} * \left(1 - FC_{long-short}\right),$$

with $FC_{long-short} = 2^{-\Delta\Delta Ct}$ calculated in the usual method.

## Relative mtDNA copy number

qPCR assays for nuclear and mitochondrial genes were performed using total DNA as a template. Relative mtDNA was calculated as follows:

mt:nuc DNA copy number $= 2 * 2^{\Delta Ct}$, where $\Delta Ct = \ Ct_{nuclear \ gene} - Ct_{mitochondrial \ gene}$.

## Immunoblots

Protein extraction was performed on cell pellets using RIPA lysis buffer, or on snap-frozen tissue using RIPA lysis buffer and a bead homogenizer. Protein lysates were fractionated using NuPage Bis-Tris polyacrylamide gels (20 μg total protein per well) and transferred to PVDF membranes. Antibodies are listed in *Table 3*.

## SA-βGal assay

Adherent cells were fixed and stained for SA-βGal via a Senescence Detection kit (Abcam, ab65351) per manufacturer's instructions. Representative images were used to calculate the fraction of SA-βGal-positive cells, scored by an observer blinded to treatment condition.

## Live cell staining

Adherent cells were stained using the TMRE-Mitochondrial Membrane Potential Assay Kit (Abcam, ab113852) per manufacturer's instructions. Select wells were pre-treated with 20 μM FCCP, a decoupling agent, as a positive control for 20 min before TMRE and Hoechst counterstain was applied. Live JAR cells were analyzed by the Celigo Image Cytometer (Nexcelom BioScience) for fluorescence quantification and gating. Adherent live JAR cells were stained with MitoSOX Red Mitochondrial Superoxide Indicator kit (Invitrogen) and analyzed using the cytometer in a similar manner.

## Cell counting

JAR cells were cultured for 6 days±cobalt chloride exposure prior to cell counting. On day 6, cells from each condition were replated in a 96-well plate in CoCl$_2$-free media at a density of 5×10$^3$ cells per well. Celigo Image Cytometer was used for automated cell counting (total cells per well) at 24 hr intervals.

## RNA-Seq

Following 6 days of treatment with $CoCl_2$ 100 µM versus control media, total RNA was isolated from JAR cells as described above. Extracted RNA (300 ng) was treated with NEBNext rRNA Depletion Kit v2 (E7400X) and cDNA was generated using random hexamer priming and Maxima H Minus Reverse Transcriptase. cDNA was converted into double-stranded cDNA using NEBNext mRNA Second Strand Synthesis Module (E6111L) and sequencing libraries were generated by tagmentation using Nextera XT DNA Library Preparation Kit (Illumina FC-131) with 12 cycles of PCR amplification. Sequencing libraries were analyzed by Qubit and Agilent Bioanalyzer, pooled at a final concentration of 1.2 pM, and sequenced on an Illumina NextSeq500 instrument 36 × 8 × 36 read structure.

## Transcriptomics analysis

Sequencing reads were demultiplexed and trimmed for adapters using bcl2fastq (v2.20.0). Secondary adapter trimming, NextSeq/Poly(G) tail trimming, and read filtering were performed using fastp (v0.20.0) (*Chen et al., 2018*) low-quality reads and reads shorter than 18 nt after trimming were removed from the read pool. Salmon (v1.1.0) (*Patro et al., 2017*) was used to simultaneously map and quantify reads to transcripts in the GENCODE 33 genome annotation of the GRCh38/hg38 human assembly. Salmon was run using full selective alignment, with sequence-specific and fragment GC-bias correction turned on (`--seqBias` and `--gcBias` options, respectively). Transcript abundances were collated and summarized to gene abundances using the tximport package for R (*Soneson et al., 2015*). Normalization and differential expression analysis were performed using edgeR (*Robinson et al., 2010*; *Chen et al., 2016*). For differential gene expression analysis, genes were considered significant if they passed an FC cutoff of log2FC >1 and an FDR cutoff of FDR <0.05. Functional enrichment analyses were performed using g:Profiler (*Raudvere et al., 2019*). WGCNA (*Langfelder and Horvath, 2008*) was performed after removing genes with low expression as previously described (*Bentsen et al., 2020*).

## Statistics

Student's two-tailed unpaired t-test, ordinary one-way, and two-way ANOVA statistical tests were applied as indicated to compare biological replicates in each experiment. Data were excluded as outliers using the interquartile range method (lower limit = first quartile – 1.5× IQR; upper limit = third quartile + 1.5× IQR).

## Study approval

All animal experiments were approved by the Beth Israel Deaconess Medical Center Institutional Animal Care and Use Committee (protocol #008-2022). Human placental specimens and data were biobanked and accessed under protocols approved by the Beth Israel Deaconess Medical Center Institutional Review Board, and written informed consent was obtained before subject participation (protocols #2008P000061, 2020P000997, 2021P000897).

# Acknowledgements

We thank Dr. Jennifer Condon of Wayne State University School of Medicine for generously providing the hTERT-HM cell line. We thank Saira Salahuddin for maintaining the placenta biorepository and helping access the study samples and accompanying clinical data. Figures were created using Biorender.com. Research reported in this publication was supported by the Institute of General Medical Sciences of the National Institutes of Health under award GM007592-41 and the Foundation for Anesthesia Education and Research (FAER) (EJC).

## Additional information

### Funding

| Funder | Grant reference number | Author |
|---|---|---|
| Foundation for Anesthesia Education and Research | Mentored Research Training Grant | Erin J Ciampa |
| National Institutes of Health | T32-GM007592 | Erin J Ciampa |
| National Institutes of Health | R35-HL139424 | Samir M Parikh |

The funders had no role in study design, data collection and interpretation, or the decision to submit the work for publication.

### Author contributions
Erin J Ciampa, Conceptualization, Formal analysis, Investigation, Methodology, Writing – original draft; Padraich Flahardy, Formal analysis, Investigation; Harini Srinivasan, Christopher Jacobs, Data curation, Formal analysis; Linus Tsai, Data curation, Formal analysis, Supervision; S Ananth Karumanchi, Samir M Parikh, Conceptualization, Resources, Supervision

### Author ORCIDs
Erin J Ciampa ![ORCID] https://orcid.org/0000-0002-6153-1994
Linus Tsai ![ORCID] http://orcid.org/0000-0002-0134-6949
S Ananth Karumanchi ![ORCID] http://orcid.org/0000-0002-2281-6831
Samir M Parikh ![ORCID] http://orcid.org/0000-0002-1038-981X

### Ethics
Human subjects: Human placental specimens and data were biobanked and accessed under protocols approved by the Beth Israel Deaconess Medical Center Institutional Review Board, and written informed consent was obtained before subject participation (protocols #2008P000061, 2020P000997, 2021P000897).

All animal experiments were approved by the Beth Israel Deaconess Medical Center Institutional Animal Care and Use Committee (protocol #008-2022).

Reviewer #1 (Public Review): https://doi.org/10.7554/eLife.85597.3.sa1
Reviewer #2 (Public Review): https://doi.org/10.7554/eLife.85597.3.sa2
Reviewer #3 (Public Review): https://doi.org/10.7554/eLife.85597.3.sa3
Author Response: https://doi.org/10.7554/eLife.85597.3.sa4

## Additional files

### Supplementary files
• MDAR checklist

### Data availability
Statistics accompanying WGCNA (*Figure 2*) are accessible through Mendeley Data, doi: https://doi.org/10.17632/g6vrw9jjn4.1. The RNA Seq dataset has been deposited in NCBI's Gene Expression Omnibus21 and is accessible through GEO Series accession number GSE199278 (https://www.ncbi.nlm.nih.gov/geo/query/acc.cgi?acc=GSE199278).

The following datasets were generated:

| Author(s) | Year | Dataset title | Dataset URL | Database and Identifier |
|---|---|---|---|---|
| Ciampa EJ, Parikh SM | 2022 | Hypoxia-inducible factor 1 signaling drives placental aging and can elicit inflammatory changes in uterine myocytes | https://www.ncbi.nlm.nih.gov/geo/query/acc.cgi?acc=GSE199278 | NCBI Gene Expression Omnibus, GSE199278 |
| Ciampa EJ | 2022 | WGCNA supplement | https://doi.org/10.17632/g6vrw9jjn4.1 | Mendeley Data, 10.17632/g6vrw9jjn4.1 |

The following previously published dataset was used:

| Author(s) | Year | Dataset title | Dataset URL | Database and Identifier |
|---|---|---|---|---|
| Knox K, Baker JC | 2008 | Expression data from developing mouse placenta | https://www.ncbi.nlm.nih.gov/geo/query/acc.cgi?acc=GSE11224 | NCBI Gene Expression Omnibus, GSE11224 |

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
