## [Editor Report · eLife assessment]

This **valuable** study provides insights into mechanisms of placental aging and its relationship to labor initiation. The authors provide **solid** evidence and have thoroughly investigated the molecular characteristics of normal placental aging using in vivo and in vitro model systems and human placental tissue analysis to corroborate their findings. This work contributes to existing work in placental aging and preterm birth and will be of interest to reproductive scientists.

---

## [Referee Report · Reviewer #1 (Public Review)]

Ciampa et al. investigated the role of the hypoxia-inducible factor 1 (HIF-1) pathway in placental aging. They performed transcriptomic analysis of prior data of placental gene expression over serial timepoints throughout gestation in a mouse model and identified increased expression of senescence and HIF-1 pathways and decreased expression of cell cycle and mitochondrial transcripts with advancing gestational age. These findings were confirmed by RT-PCR, Western blot, and mitochondrial assessment from mouse placental tissues from late gestation time points. Studies of human placental samples at similar late gestational ages showed similar trends in increased HIF-1 targets and decreased mitochondrial abundance with increasing gestation, but were not significantly significant due to the limited availability of uncomplicated preterm placenta samples. The authors demonstrated that stabilization of HIF-1 in vitro using primary trophoblasts and choriocarcinoma cell lines recapitulated the gene and mitochondrial dysfunction seen in the placental tissues and were consistent with senescence. Interestingly, cell-conditioned media from HIF-1 stabilized placenta cell lines induced myometrial cell contractions in vitro and correspondingly, induction of HIF-1 in pregnant mice was associated with preterm labor in vivo. These data support the role of the HIF-1 pathway in the process of placental senescence with increasing gestational age and highlight this pathway as a potentially important contributor to gestational length and a potential target for therapeutics to reduce preterm birth.

Overall, the conclusions of this study are mostly well supported by the data. The concept of placental aging has been controversial, with several prior studies with conflicting viewpoints on whether placental aging occurs at all, is a normal process during gestation, or rather only a pathologic phenomenon in abnormal pregnancies. This has been rather difficult to study given the difficulty of obtaining serial placental samples in late gestation. The authors used both a mouse model of serial placental sampling and human placental samples obtained at preterm, but non-pathologic deliveries, which is an impressive accomplishment as it provides insight into a previously poorly understood timepoint of pregnancy. The data clearly demonstrate changes in the HIF-1 pathway and cellular senescence at increasing gestational ages in the third trimester, which is consistent with the process of aging in other tissues.

Weaknesses of this study are that although the authors attribute alterations in HIF-1 pathways in advanced gestation to hypoxia, there are no experiments directly assessing whether the changes in HIF-1 pathways are due to hypoxia in either in vitro or in vivo experiments. HIF-1 has both oxygen-dependent and oxygen-independent regulation, so it is unclear which pathways contribute to placental HIF-1 activity during late gestation, especially since the third-trimester placenta is exposed to significantly higher oxygen levels compared to the early pregnancy environment. Additionally, the placenta is in close proximity to the maternal decidua, which consists of immune and stromal cells, which are also significantly affected by HIF-1. Although the in vitro experimental data in this study demonstrate that HIF-1 induction leads to a placenta senescence phenotype, it is unclear whether the in vivo treatment with HIF-1 induction acts directly on the placenta or rather on uterine myometrium or decidua, which could also contribute to the initiation of preterm labor.

---

## [Referee Report · Reviewer #2 (Public Review)]

The authors sought to characterize normal placental aging to better understand how the molecular and cellular events that trigger the labor process. An understanding of these mechanisms would not only provide insight into term labor, but also potential triggers of preterm labor, a common pregnancy complication with no effective intervention. Using bulk transcriptomic analysis of mouse and human placenta at different gestational timepoints, the authors determined that stabilization of HIF-1 signaling accompanied by mitochondrial dysfunction and cellular senescence are molecular signatures of term placenta. They also used in vitro trophoblast (choriocarcinoma) and a uterine myocyte culture system to further validate their findings. Lastly, using chemically induced HIF-1 induction in vivo in mice, the authors showed that stabilization of HIF-1 protein in the placenta reduced the gestational length significantly.

The major strength of this study is the use of multiple model systems to address the question at hand. The consistency of findings between mouse and human placenta, and the validation of mechanisms in vitro and in vivo modeling are strong support for their conclusions. The rationale for studying the term placentas to understand the abnormal process of preterm birth is clearly explained. Although the idea that hypoxic stress and placental senescence are triggers for labor is not novel, the comprehensiveness of the approach and idea to study the normal aging process are appreciated.

There are some areas of the manuscript that lack clarity and weaknesses in the methodology worth noting. The rationale for focusing on senescence and HIF-1 is not clearly given that other pathways were more significantly altered in the WGCNA analysis. The placental gene expression data were from bulk transcriptomic analyses, yet the authors do not explicitly discuss the limitations of this approach. Although the reader can assume that the authors attribute the mRNA signature of aging to trophoblasts - of which, there are different types - clarification regarding their interpretation of the data and the relevant cell types would strengthen the paper. Additionally, while the inclusion of human placenta data is a major strength, the differences between mouse and human placental structure and cell types make highlighting the specific cells of interest even more important; although there are correlations between mouse and human placenta, there are also many differences, and the comparison is further limited when considering the whole placenta rather than specific cell populations.

Additional details regarding methods and the reasons for choosing certain readouts are needed. Trophoblasts are sensitive to oxygen tension which varies according to gestational age, and it is unclear if this variable was taken into consideration in this study. Many of the cellular processes examined are well characterized in the literature yet the rationale for choosing certain markers is unclear (e.g., Glb1 for senescence; the transcripts selected as representative of the senescence-associated secretory phenotype; mtDNA lesion rate).

Overall, the findings presented are a valuable contribution to the field. The authors provide a thoughtful discussion that places their findings in the context of current literature and poses interesting questions for future pursuit. Their efforts to be comprehensive in the characterization of placental aging is a major strength; few placental studies attempt to integrate mouse and human data to this extent, and the validation and presentation of a potential mechanism by which fetal trophoblasts signal to maternal uterine myocytes are exciting. Nevertheless, a clear discussion of the methodologic limitations of the study would strengthen the manuscript.

---

## [Referee Report · Reviewer #3 (Public Review)]

In this study, Ciampa and colleagues demonstrate that HIF-1α activity is increased with gestation in humans and mice placentas and use several in vitro models to indicate that HIF activation in trophoblasts may release factors (yet to be identified) which promote myometrial contraction. Previous studies have linked placental factors to the preparation of the myometrium for labour (e.g. prostaglandins), but HIF-1α has not been implicated.

Weaknesses and concerns:

1. The author's rebuttal state that placentas undergo subclinical cellular aging as they reach term. Although several future studies are described to test functional deficits at the cellular level, the current manuscript does not provide convincing evidence of cellular aging. The only evidence of cellular senescence provided in both human and mouse data is the mRNA expression of a single gene associated with senescence.

2. The authors have not responded to the concern regarding CoCl2 mediating differentiation. The paragraph from a ref states that JAR cells do not respond as well as BeWOs to forskolin. However, this does not mean that JAR cells do not differentiate. This point is particularly pertinent as a quick search of their RNA-seq data shows upregulation of STB genes following CoCl2 treatment including ERVs (ERVFRD1, ERVV-1, ERVV-2, ERV3-1), CYP19A1 and OVOL1 just to name a few. If the authors' conclusion is that CoCl2 treatment did not alter trophoblast differentiation, the authors should provide additional data showing this. For example, cell fusion assays showing E-cadherin/desmoplakin staining and nuclear localization within stained boundaries.

3. The authors acknowledge the possibility of extraplacental effects of DMOG in the initiation of labour in their model, no additional evidence has been provided to support placental effects of their model. The authors also argue that although PMID 30808919 (which specifically overexpressed HIF-1a in the placenta) did not show changes in birth length, they propose that this may be due to constitutive HIF1a expression at the beginning of pregnancy. This argument is invalid since placental maldevelopment is consistently linked with several pregnancy complications including spontaneous preterm birth. If anything, perturbations in the beginning of pregnancy are more likely to lead to worse outcomes than those at the end of pregnancy.

4. Regarding induction of syncytialisation, please provide additional evidence that the cells have/have not syncytialised.

5. Lack of cohesion between experimental models. Please provide evidence that DMOG mediates similar effects on SA-β gal activity as CoCl2 in JARs.

6. Evidence of senescence and mitochondrial abundance could be strengthened by providing additional markers. E.g. only GLB1 mRNA expression is provided as evidence of senescence, and COX IV protein for mitochondrial abundance in mouse and human placentas. This point has not been addressed. Please provide at least one additional marker of senescence and mitochondrial abundance.

---

## [Author Response]

The following is the authors’ response to the original reviews.

**Reviewer #1 (Public Review):**
[…] Overall, the conclusions of this study are mostly well supported by the data. The concept of placental aging has been controversial, with several prior studies with conflicting viewpoints on whether placental aging occurs at all, is a normal process during gestation, or rather only a pathologic phenomenon in abnormal pregnancies. This has been rather difficult to study given the difficulty of obtaining serial placental samples in late gestation. The authors used both a mouse model of serial placental sampling and human placental samples obtained at preterm, but non-pathologic deliveries, which is an impressive accomplishment as it provides insight into a previously poorly understood timepoint of pregnancy. The data clearly demonstrate changes in the HIF-1 pathway and cellular senescence at increasing gestational ages in the third trimester, which is consistent with the process of aging in other tissues.Weaknesses of this study are that although the authors attribute alterations in HIF-1 pathways in advanced gestation to hypoxia, there are no experiments directly assessing whether the changes in HIF-1 pathways are due to hypoxia in either in vitro or in vivo experiments. HIF-1 has both oxygen-dependent and oxygen-independent regulation, so it is unclear which pathways contribute to placental HIF-1 activity during late gestation, especially since the third-trimester placenta is exposed to significantly higher oxygen levels compared to the early pregnancy environment. Additionally, the placenta is in close proximity to the maternal decidua, which consists of immune and stromal cells, which are also significantly affected by HIF-1. Although the in vitro experimental data in this study demonstrate that HIF-1 induction leads to a placenta senescence phenotype, it is unclear whether the in vivo treatment with HIF-1 induction acts directly on the placenta or rather on uterine myometrium or decidua, which could also contribute to the initiation of preterm labor.

We thank Reviewer #1 for the thoughtful analysis offered here. We agree that our study has not determined whether placental HIF-1 activation occurring during late gestation is due to oxygen-dependent or oxygen-independent regulation; both possibilities are outlined in paragraph 3 of the Discussion. We used a pharmacological approach in our experiments characterizing the effects of HIF-1 stabilization in trophoblasts because it allows superior command of experimental conditions, but in future studies using hypoxic growth conditions we will determine whether oxygen sensing is a critical component of the aging effects on mitochondrial abundance, metabolism, and cellular senescence in the placenta.

Reviewer #1 also appropriately highlights the possibility that extra-placental effects of DMOG may contribute to the initiation of preterm labor in our mouse model. Future studies making use of mice with placenta-specific transgenes will allow clarification of the specific contributions of placental HIF-1 signaling to labor onset.

**Reviewer #2 (Public Review):**
[…] The major strength of this study is the use of multiple model systems to address the question at hand. The consistency of findings between mouse and human placenta, and the validation of mechanisms in vitro and in vivo modeling are strong support for their conclusions. The rationale for studying the term placentas to understand the abnormal process of preterm birth is clearly explained. Although the idea that hypoxic stress and placental senescence are triggers for labor is not novel, the comprehensiveness of the approach and idea to study the normal aging process are appreciated.There are some areas of the manuscript that lack clarity and weaknesses in the methodology worth noting. The rationale for focusing on senescence and HIF-1 is not clearly given that other pathways were more significantly altered in the WGCNA analysis. The placental gene expression data were from bulk transcriptomic analyses, yet the authors do not explicitly discuss the limitations of this approach. Although the reader can assume that the authors attribute the mRNA signature of aging to trophoblasts - of which, there are different types - clarification regarding their interpretation of the data and the relevant cell types would strengthen the paper. Additionally, while the inclusion of human placenta data is a major strength, the differences between mouse and human placental structure and cell types make highlighting the specific cells of interest even more important; although there are correlations between mouse and human placenta, there are also many differences, and the comparison is further limited when considering the whole placenta rather than specific cell populations.Additional details regarding methods and the reasons for choosing certain readouts are needed. Trophoblasts are sensitive to oxygen tension which varies according to gestational age, and it is unclear if this variable was taken into consideration in this study. Many of the cellular processes examined are well characterized in the literature yet the rationale for choosing certain markers is unclear (e.g., Glb1 for senescence; the transcripts selected as representative of the senescence-associated secretory phenotype; mtDNA lesion rate).Overall, the findings presented are a valuable contribution to the field. The authors provide a thoughtful discussion that places their findings in the context of current literature and poses interesting questions for future pursuit. Their efforts to be comprehensive in the characterization of placental aging is a major strength; few placental studies attempt to integrate mouse and human data to this extent, and the validation and presentation of a potential mechanism by which fetal trophoblasts signal to maternal uterine myocytes are exciting.Nevertheless, a clear discussion of the methodologic limitations of the study would strengthen the manuscript.

We thank Reviewer #2 for careful consideration of our data and for the valuable feedback.

We chose to focus on HIF-1 signaling, mitochondrial function and abundance, and cellular senescence among the pathways that emerged from WGCNA based on our testable hypothesis that these three phenomena could be linked, with HIF-1 upstream of mitochondrial changes and cellular senescence (noted in Lines 166-169 with references to studies on aging establishing this connection in other systems). The other pathways not studied here (FOXO, AMPK, mTOR signaling) are important stress-response mediators which likely play additional key roles in the biology we have begun to describe; extensive future studies are warranted to explore this fully.

While we focused on establishing new mechanistic insights for aging in the placenta as a whole, localization of the effects described here to specific placental cell populations will be important to clarify in future studies, as is proposed in the Discussion (lines 316-319, which has been updated for emphasis). To our knowledge, no single-cell transcriptomics studies of the placenta have been published to date describing gene expression changes across advancing gestational age in healthy pregnancies, and the quantitative value of immunolocalization studies of candidate proteins in isolation would be limited.

We do not dispute the limitations of mouse placenta as an imperfect model for the human organ; we have provided parallel data from human specimens wherever possible. We agree that this will continue to be critical in future studies, especially those aiming to achieve cell-type localization of these signaling pathways.

As mentioned in the response to Reviewer #1, we utilized pharmacological HIF-1 induction in our experimental models rather than manipulation of oxygen tension but acknowledge the value of follow-up studies utilizing hypoxic growth conditions in the Discussion.

SA-b-Gal activity is a key biomarker of cellular senescence, and this is most commonly assessed histochemically. Unfortunately, detecting b-galactosidase enzyme activity was not possible in the biobanked human specimens we accessed in this study (not collected/stored in a suitable format for histochemical processing), which is why we instead quantified expression of the lysosomal enzyme b-D-galactosidase, encoded by *GLB1*, the gene responsible for SA-b-Gal activity (Lee BY *et al*. Senescence-associated β-galactosidase is lysosomal β-galactosidase. *Aging Cell* 2006 – cited in line 106). A host of other senescence markers exists, but their appearance in senescent cells depends on the cell type and underlying drivers of the senescent phenotype (reference #45), with SA-b-Gal activity among the most universal. Similarly, the specific SASP components depend on cell type and senescence stimulus; we selected the markers in Figure 5H based on their previously established roles as mediators of placental signaling. As noted in the text (lines 120-121 with references to the relevant literature), mtDNA damage has previously been implicated as a driver of age-related loss-of-function in other tissues, which led us to explore whether mtDNA damage accompanies the other signs of mitochondrial dysfunction and dysregulation that were emerging in our data.

**Reviewer #3 (Public Review):**
In this study, Ciampa and colleagues demonstrate that HIF-1α activity is increased with gestation in humans and mice placentas and use several in vitro models to indicate that HIF activation in trophoblasts may release factors (yet to be identified) which promote myometrial contraction. Previous studies have linked placental factors to the preparation of the myometrium for labour (e.g. prostaglandins), but HIF-1α has not been implicated. Due to several issues regarding the experimental design, the results do not currently support the conclusions.Major concerns:1. The hypothesis states that placental aging promotes parturition via HIF-1a activation, the study does not provide any evidence of an aged placenta. Aging is considered a progressive and irreversible loss of functional capacity, inability to maintain homeostasis, and decreased ability to repair the damage. The placenta retains all these abilities throughout pregnancy [PMID: 9462184], and there's no evidence that the placenta functionally declines between 35-39 weeks, otherwise, it wouldn't be able to support fetal development. However, there is evidence of a functional decline in post-term placentas (i.e. >40 weeks in humans) but the authors compare preterm placentas with E17.5 mice placentas or 39-week human placentas, both these gestational periods are prior to the onset of parturition in most pregnancies (human = 40wkGA, mice=E18.5).

We thank Reviewer #3 for careful consideration of our manuscript and the thoughtful feedback.

Our stance that the placenta ages across its normal lifespan is based on the appearance of cellular senescence as an emerging pathway in latter gestational timepoints in the WGCNA, with subsequent validation of cellular senescence markers accumulating in placental samples from the advanced gestational age cohort. Although functional deficits stemming from the appearance of cellular senescence late in pregnancy may not be appreciable at the whole-system level until post-dates, we propose that the subclinical cellular aging that we have detected even before labor onset may be relevant in the setting of a “second hit” stressor — eg, impaired ability to maintain homeostasis, repair damage.

Future studies will examine functional deficits at the cellular level in response to HIF-1 stabilization (eg. Seahorse assay) and in early- versus late-gestational age primary cells. We hypothesize such studies will reveal impaired resistance to metabolic stressors in the senescent phenotype. Further, there will be value in exploring the impact of senolytics in restoring function to aged tissue.

In both mouse and human, our selection of placentas that had not yet been exposed to spontaneous labor was deliberate, in order to avoid confounding from the inflammatory effects of labor and delivery itself (due to large swings in perfusion pressure and local ischemia-reperfusion events).

1. While the authors provide evidence that HIF-1α activity increases in both the human and mice placenta as gestation progresses, the mechanistic link between placental HIF-1α and parturition is not strongly supported. For example, most of the evidence is based on in vitro studies showing that conditioned media from trophoblasts treated with CoCl2 increased the contraction of myometrial cells. The specific factor responsible was not identified but the authors allude to pro- inflammatory factors such as cytokines. It was therefore interesting to note that the conditioned media had undergone a filtration step that removes all substances >10kDa, which includes the majority of cytokines and hormones.

We appreciate the opportunity to clarify that in the filtration step, we *retained the >10 kDa fraction*, allowing us to clear CoCl2 itself among other <10kDa molecules. A 10kDa cutoff was chosen to allow for retention of cytokines including those previously implicated as signals that can promote contractility in uterine myocytes. As mentioned in the discussion, future studies will aim to identify specific factors within the secretome that are necessary and sufficient to induce the contractile changes.

1. An alternative explanation is that CoCl2 treatment-induced trophoblast differentiation and the effects on myometrial contraction may be related to differences in secreted factors produced by cytotrophoblasts versus syncytiotrophoblast. Although JAR cells do not spontaneously differentiate, they can be induced to syncytialise upon cAMP stimulation. Ref 39 the authors cite shows this. Indeed, the morphology of the cells in Fig5F that were exposed to CoCl2 indicates that they may be syncytialised. Syncytialised trophoblasts also express markers of senescence including increased SA-β-gal activity and reductions in mitochondrial activity.

The following is taken from Reference 39, final paragraph:

For instance, among the tested cell lines the choriocarcinoma cell line BeWo is best suited for studies on syncy8al fusion. However, ACH-3P, JAR and Jeg-3 cells react to forskolin treatment with elevated levels of hCG they do not form syncy8a73 and are therefore poor models for syncy8aliza8on over a period of 7

days.

1. The in vivo experiment showing reduced gestation length in pregnant mice receiving DMOG injection is interesting. However, we cannot exclude the effects of DMOG on non-placental tissues (both maternal and fetal) or the non-specific effects of DMOG (i.e. HIF-1α independent). Furthermore, previous studies using a more direct approach to alter HIF-1α activity in the placenta using trophoblast-specific overexpression of HIF-1α in mice do not lead to changes in gestation length [PMID: 30808910].

As stated in the response to Reviewer #1, we acknowledge the possibility that extra-placental effects of DMOG may contribute to the initiation of preterm labor in our mouse model. Future studies making use of mice with placenta-specific transgenes will allow clarification of the specific contributions of placental HIF-1 signaling to labor onset.

Regarding PMID 30808919, as noted in our Discussion (lines 326-335), an important distinction is that the referenced paper studied effects of trophoblast- specific expression of a constitutively active HIF1 from the beginning of pregnancy, and their findings highlight markedly abnormal placental development in that context. By contrast, we describe effects of HIF-1 stabilization late in pregnancy in a normally developed placenta.

1. Lack of appropriate experimental models. E.g. JAR choriocarcinomas are not an ideal model of the human trophoblast as they are malignant. Much better models are available e.g. primary human trophoblasts from term placentas or human trophoblast stem cells from first-trimester placentas. Similarly, the mouse model is also not specific as discussed above.

We agree with the Reviewer that the JAR cell line has important differences from human trophoblasts, nonetheless as stated in the Results section (Lines 181-184) they were used in order to model long-term exposure to HIF-1 induction without underlying syncytialization confounding the findings, as would be the case with primary cells.

1. Lack of cohesion between the different experimental models. E.g. CoCl2 was used to induce hypoxia/HIF1α in mouse TBs, but DMOG was used in vivo in mice. SA-β Gal staining was carried out in cells but not in mouse or human tissues.

We used two distinct prolyl hydroxylase inhibitors (CoCl2 and DMOG) in our in vitro studies (Figures 4, 5, and 5 Supplement) to demonstrate reproducibility across models and to help attribute the effects to HIF-1 stabilization rather than off-target effects. DMOG was chosen for the in vivo studies because of its prior use in mice.

As mentioned in response to reviewer 2, detecting b-galactosidase enzyme activity was not possible in the biobanked human specimens we accessed in this study (not collected/stored in a suitable format for histochemical processing), which is why we instead quantified expression of the lysosomal enzyme b-D- galactosidase, encoded by *GLB1*, the gene responsible for SA-b-Gal activity (Lee BY *et al*. Senescence-associated β-galactosidase is lysosomal β-galactosidase. *Aging Cell* 2006 – cited in line 106).

1. Evidence of senescence and mitochondrial abundance could be strengthened by providing additional markers. E.g. only GLB1 mRNA expression is provided as evidence of senescence, and COX IV protein for mitochondrial abundance in mouse and human placentas.

As mentioned in response to Reviewer 2, the appearance of other senescence markers depends on the cell type and underlying drivers of the senescent phenotype (reference #45), with SA-b-Gal activity among the most universal. Future studies will further probe which markers accompany cellular senescence in aging placenta to define the senescence phenotype in this setting.

1. Given that the main goal of this study was to investigate the role of hypoxia, hypoxia (i.e. low oxygen) was never directly induced and the results were based on chemical inducers of HIF-1α which have multiple off-target effects.

As mentioned in response to Reviewer 1, we agree that our study has not determined whether placental HIF-1 activation occurring during late gestation is due to oxygen-dependent or oxygen-independent regulation; both possibilities are outlined in paragraph 3 of the Discussion. We used a pharmacological approach in our foundational experiments characterizing the effects of HIF-1 stabilization in trophoblasts because it allows superior command of experimental conditions, but in future studies using hypoxic growth conditions we will determine whether oxygen sensing is a critical component of the aging effects on mitochondrial abundance, metabolism, and cellular senescence in the placenta. We are encouraged by the consistency of the senescence phenotype in JAR cells following administration of two distinct prolyl hydroxylase inhibitors, CoCl2 and DMOG, suggesting that the effects seen are more likely attributable to HIF-1 stabilization (versus off-target effects).

**Reviewer #1 (Recommendations For The Authors):**
This is a very interesting and well-written study that supports the concept of placental aging using a combination of a mouse model, in vitro cell lines, and human placental samples.Overall this is an important contribution to our current understanding of placental biology highlighting the role of the HIF-1 pathway and merits publication.This study would be strengthened by the following addition:- As stated in the Public Review, the authors attribute HIF-1 induction at increased gestation to hypoxia, however, this has not been demonstrated experimentally and HIF-1 has both O2-dependent and independent regulation. The authors could perform an in vitro culture of primary placental cells or JAR cells under hypoxic conditions and assess the HIF-1 pathway/mitochondria activity to provide support for a hypoxia-dependent mechanism.

We thank Reviewer #1 for the thoughtful analysis offered here. We agree that our study has not determined whether placental HIF-1 activation occurring during late gestation is due to oxygen-dependent or oxygen-independent regulation; both possibilities are outlined in paragraph 3 of the Discussion. We used a pharmacological approach to characterize effects of HIF-1 stabilization in trophoblasts because it allows superior command of experimental conditions, but in future studies using hypoxic growth conditions we will determine whether oxygen sensing is a critical component of the aging effects on mitochondrial abundance, metabolism, and cellular senescence in the placenta.

**Reviewer #2 (Recommendations For The Authors):**
Major comments:1. The rationale for the pursuit of HIF-1 and cellular senescence after initial WGCNA was weakly supported, though this avenue led to interesting and impactful results. The text could provide a stronger rationale for pursuing these pathways as opposed to the top- upregulated and downregulated pathways, perhaps by emphasizing previously published work in the field.

We thank Reviewer #2 for careful consideration of our data and for the valuable feedback.

We chose to focus on HIF-1 signaling, mitochondrial function and abundance, and cellular senescence among the pathways that emerged from WGCNA based on our testable hypothesis that these three phenomena could be linked, with HIF-1 upstream of mitochondrial changes and cellular senescence (noted in Lines 166-169 with references to studies establishing this connection in other systems). The other pathways not studied here (FOXO, AMPK, mTOR signaling) are important stress-response mediators which likely play additional key roles in the biology we have begun to describe; extensive future studies are warranted to explore this fully.

1. Validation of the gene expression data with placental histology and immunolocalization of proteins of interest would bolster the study by identifying the relevant cell types and showing changes in protein levels over time. Additionally, single-cell RNA-seq data from mouse and human placenta are available. Exploration of these published datasets would also be interesting.

While we focused on establishing new mechanistic insights for aging in the placenta as a whole, localization of the effects described here to specific placental cell populations will be important to clarify in future studies, as is proposed in the Discussion (lines 316-319, which has been updated for emphasis). To our knowledge, no single-cell transcriptomics studies of the placenta have been published to date describing gene expression across advancing gestational age timepoints, and the value of single timepoint “snapshots” that exist in the literature is limited for the purpose of validating the aging mechanisms we have proposed here.

3. In Figure 2, all of the data have a gestational age-dependent trend except for Fig 2F where the mtDNA lesion rate drops at e15.5. What is the authors' interpretation of these results?

A testable hypothesis to explain this result is that as mtDNA damage begins to accumulate, cells are initially able to respond via mitophagy, removing those mitochondria with damaged DNA (e15.5), until that response is overwhelmed, allowing the detectable mtDNA lesion rate to spike at e17.5.

1. In paragraph three of the results, the authors conclude that there is an accumulation of ROS stress, yet there is no direct measurement of ROS. Measuring ROS directly in this setting would strengthen this conclusion (similar to what is done in Figure 5E).

We interpreted the accumulation of mtDNA damage as a marker of ROS stress, but the Reviewer correctly points out that we did not measure ROS directly in this model. We have updated the language (line 126) to be more accurate.

1. There is a discrepancy in the length of CoCl2 treatment in primary trophoblasts vs. JAR cells (48 hours vs. 6 days). Treatment with DMOG in JAR cells also differed (4 days). Do the authors have any evidence that longer vs. shorter stabilization of HIF-1 has secondary effects in these cells that could affect the results of the study?

We preliminarily explored the timecourse of the effects of HIF-1 stabilization in JAR cells, as shown in Fig 5 – Supp 1, and also found that the decline in mt abundance precedes the appearance of senescence markers (data not shown). JAR cells are a much better model for exploring effects of chronic exposure to HIF-1 stabilization because they do not syncytialize as primary trophoblasts do. We limited our studies in primary cells for this reason to a 48h- timepoint, because the effects of syncytialization would confound longer studies. With the aim of simply validating our CoCl2 findings with a separate prolyl hydroxylase inhibitor, we picked an intermediate timepoint for convenience. The reviewer correctly pinpoints the value of future studies that further dissect the kinetics of these phenomena, which could also potentially identify at which phases the effects are reversible.

1. The authors evaluated mitochondrial effects in a time course experiment (Figure 5 Supplement 1) and found that the effects of HIF-1 stabilization emerged after three days of treatment, but no such experiment was conducted to determine the timing of senescence with SA-βGal. It would be interesting to correlate the mitochondrial effects and onset of senescence caused by HIF-1 stabilization.

In future studies we will continue to explore the relative dynamics of HIF1 stabilization vs mitochondrial effects and senescence. In doing so it will be important to explore other markers of senescence; while SAbGal is the most universal senescence marker, others (such as p16 or p21 induction), if present, may lend themselves to more precise quantification and a clearer definition of senescence “start time”.

7. IL-1β is used in experiments testing the effect of JAR-conditioned media on uterine myocytes. The conclusion of this experiment is that conditioned media from JAR cells treated with CoCl2, but not from untreated JAR cells, results in myocyte contraction (Figure 6E) and expression of contraction-associated genes (Figure 6A-D). Although the figure shows that IL-1β + conditioned media increases expression of these genes compared to IL- 1β alone, an added control condition where conditioned media is used in the absence of IL- 1β would underscore this conclusion and show whether the components in the conditioned media are sufficient to induce gene expression and contraction. There is also no justification for the 10 kDa cutoff in this experiment.

We did test whether conditioned media could induce contractile changes in myocytes in the absence of IL-1b co-stimulation, and indeed found that the CoCl2-stimulated conditioned media does elicit this effect on its own. We eliminated these conditions from the published figure in an aim to limit its complexity, but present them here (*, p< 0.05 vs no treatment):

**Author response image 1. sa4fig1:** 

The filtration step was implemented to concentrate the conditioned media prior to applying it to the myocytes. A 10kDa cutoff was chosen to ensure retention of most cytokines, especially those previously implicated in contractile switching of uterine myocytes (eg. IL1b, IL1a, TNFa each approximately 18 kDa, IL6 approximately 21 kDa). The filtration and wash steps also ensured clearance of CoCl2 out of the conditioned media before it was applied to myocytes.

1. Figure 7 shows the use of DMOG in vivo to stabilize HIF-1, which induces preterm labor. Is there a way to inhibit HIF-1 signaling downstream to show that preterm labor in vivo is specifically due to HIF-1 stabilization and not an off-target effect of DMOG? Rescue experiments either in vitro or in DMOG-treated mice using HIF-1s inhibitors would be very compelling although we recognize these may not be feasible. Regardless, a comment on the translational impact of this study and the potential of targeting the HIF pathway to treat or prevent SPTB should be considered.

There is considerable research into HIF inhibitors as cancer therapeutics (and FDA approval of a HIF2a inhibitor, belzutifan, for von Hippel Lindau disease). Future studies into the ability of HIF-1 inhibitors to rescue preterm labor are certainly of interest, though translational potential may be limited by systemic toxicity unless a targeted placenta-specific delivery system can be achieved. Genetic approaches using placenta-specific knockout might also be useful, particularly if conditional knockout can be achieved to limit the effects on HIF-1 signaling to late pregnancy, after placental development is complete.

9. The effect of JAR-conditioned media on uterine myocytes is very interesting. The authors might consider additional discussion of what the putative mediators are and what is suggested in the preterm birth literature (e.g., Sheller-Miller, PMID: 30679631). Assessment of other SASP factors in using ELISA, e.g., would strengthen the study, or at least a rationale for the genes evaluated.

We agree that follow-up studies should be done to identify which components of the secretome are key for mediating the contractile effect in myocytes, as noted in the Discussion (Lines 271-273), now updated for emphasis and with the suggested references.

Additional minor comments:1. For Figure 1A, without reading the figure legend it is unclear that the vertical color graph represents different gene clusters; consider labeling the y-axis with 'Gene clusters.' Also, blue and turquoise clusters could be labeled as "upregulated" or "downregulated" for simplicity and clarity.

Updated, thank you for the suggestions.

1. For mRNA expression wherever relevant, state in the figure legends and main text the method used (i.e., qPCR) and what the reference timepoint and normalization strategy was. For instance, in Figure 2 (and supplement 1), we were of the impression that the e15.5 and e17.5 values were normalized to e13.5.

Updated, thank you for the suggestions.

1. For Figure 5, can the authors explain in the main text what is Mtsox and how is it a marker for mitochondrial depolarization? In 5E, it would be helpful to mention what is TMRE and FCCP are and how it measures mitochondrial ROS.

Updated, thank you for the suggestions.

1. Figure 5 Supplement 2 and Figure 5 Supplement 3 appear to be missing labels indicating black vs. blue vs. red datasets.

Updated, thank you for the suggestion.

1. Figure 7c, what is the n in each group?

Gestational length data in Figures 7c and 7d each reflect the same n=8 mice per group.

1. Minor edits are needed for inconsistent use of terms (pre-term vs. preterm, for example) and grammar.

Updated, thank you for the suggestion.

Suggested additions to the Methods section to improve reproducibility:1. Include more detail re: cell culture conditions, including % oxygen.

Updated, thank you.

1. Collagen lattice contraction assay - include details on how measurements of collagen discs were performed. Was this automated?

Updated, thank you.

1. Immunoblots. Details, such as the amount of protein loaded, gel composition, protein extraction method, etc., would be helpful.

Updated, thank you.

**Reviewer #3 (Recommendations For The Authors):**
Minor comments:1. It is unclear why 2-way ANOVA was performed in figure 3 when there are only 2 groups under comparison: <35 wks vs >39 wks

As in Figure 2D, multiple genes are analyzed together in Figure 3A using 2-way ANOVA with the two factors being (1) gestational age and (2) individual gene targets (GLB1, HK2, GLUT1). This approach allows us to define the combined effect of gestational age on expression level of all of the genes whose expression is increasing.

1. Scale bars missing in some figures - Fig4E, Fig 5D, 5F, Fig5 - Suppl 3C.

Scale bars were not captured with the original images; we regret this omission.